# The geometric basis of epithelial convergent extension

**Fridtjof Brauns[1]\*[†], Nikolas H Claussen[2][†], Matthew F Lefebvre[2], Eric F Wieschaus[3,4], Boris I Shraiman[1,2]\***

[1]Kavli Institute for Theoretical Physics, University of California, Santa Barbara, Santa Barbara, United States; [2]Department of Physics, University of California, Santa Barbara, Santa Barbara, United States; [3]Department of Molecular Biology, Princeton University, Princeton, United States; [4]The Lewis-Sigler Institute for Integrative Genomics, Princeton University, Princeton, United States

**\*For correspondence:**
fbrauns@kitp.ucsb.edu (FB);
shraiman@kitp.ucsb.edu (BIS)

[†]These authors contributed equally to this work

**Competing interest:** The authors declare that no competing interests exist.

## eLife assessment

This **important** study analyzes in an original way how tension pattern dynamics can reveal the contribution of active versus passive intercalation during tissue elongation. The authors develop a **compelling**, elegant analytical framework (isogonal tension decomposition) to disentangle the passive (adjacent tissues pulling) and active (local tension anisotropy) contributions to intercalation events. This allows the generation of global maps of tissue mechanics that will be extremely helpful in the field of biomechanics.

**Abstract** Shape changes of epithelia during animal development, such as convergent extension, are achieved through the concerted mechanical activity of individual cells. While much is known about the corresponding large-scale tissue flow and its genetic drivers, fundamental questions regarding local control of contractile activity on the cellular scale and its embryo-scale coordination remain open. To address these questions, we develop a quantitative, model-based analysis framework to relate cell geometry to local tension in recently obtained time-lapse imaging data of gastrulating *Drosophila* embryos. This analysis systematically decomposes cell shape changes and T1 rearrangements into internally driven, active, and externally driven, passive, contributions. Our analysis provides evidence that germ band extension is driven by active T1 processes that self-organize through positive feedback acting on tensions. More generally, our findings suggest that epithelial convergent extension results from the controlled transformation of internal force balance geometry which combines the effects of bottom-up local self-organization with the top-down, embryo-scale regulation by gene expression.

## Introduction

Tissue elongation is a basic element of morphogenesis. Tissues can elongate by oriented cell divisions and growth (*Mao et al., 2011*; *Gillies and Cabernard, 2011*), or via local rearrangements of cells. The latter happens during epithelial 'convergent extension' – a common motif of early development and organogenesis in many organisms – where an epithelium elongates along one axis while contracting along the perpendicular direction (*Huebner and Wallingford, 2018*). Epithelial convergent extension exemplifies the important role of cell and tissue mechanics in morphogenetic processes, which must be studied alongside developmental genetics and signaling. The fundamental question of developmental mechanics is how force generation is coordinated across cells to produce a coherent morphogenetic outcome.

The *Drosophila* embryo is one of the best-established models of developmental mechanics as it is ideal for live imaging and offers an extensive set of genetic tools (*Hales et al., 2015*). Progress in live imaging and computational image analysis has produced remarkably quantitative data (*Krzic et al., 2012*; *Mitchell et al., 2022*). In this work, we will take advantage of a previously published dataset which we will reanalyze and use as a test-bed for theory development. *Figure 1* recapitulates the basic quantitative features of germ band elongation (GBE) during *Drosophila* embryogenesis based on the light-sheet imaging data from *Stern et al., 2022*; *Mitchell et al., 2022* which, thanks to surface extraction (*Heemskerk and Streichan, 2015*) and cell segmentation and tracking (*Stern et al., 2022*), provide a global picture of tissue dynamics with cellular resolution. During *Drosophila* gastrulation, the embryonic blastoderm (an epithelial monolayer of about 6000 cells on the surface of the embryo) undergoes dramatic deformation that changes tissue topology and gives rise to the three germ layers (*Gilbert and Barresi, 2016*). Gastrulation starts (see *Figure 1A*) with the formation of the ventral furrow (VF) which initiates the internalization of mesoderm, followed immediately by germ band extension (GBE), which involves convergent extension of the lateral ectoderm (or germ band) and the flow of tissue from the ventral onto the dorsal side of the embryo (*Leptin, 1995*; *Martin et al., 2009*; *Martin, 2020*; see *Video 1*). Concomitant with GBE, the posterior midgut (PMG) moves from the posterior pole towards the anterior on the dorsal side of the embryo and invaginates progressively as it moves.

VF and PMG invagination and GBE have been extensively studied, leading to the identification of relevant developmental patterning genes (*Irvine and Wieschaus, 1994*; *Leptin, 1995*; *Martin et al., 2009*; *Martin, 2020*). Live imaging has also uncovered the pertinent cell behavior during GBE, namely intercalation of neighboring cells in the lateral ectoderm (*Irvine and Wieschaus, 1994*; *Zallen and Wieschaus, 2004*; *Bertet et al., 2004*). The elementary step of this cell rearrangement process is called a T1 transition. It involves a quartet of cells where two cells lose contact and the two other cells gain contact (see *Figure 1B*, bottom). The role of intercalations is highlighted by a 'tissue tectonics' (*Blanchard et al., 2009*) analysis, which decomposes the tissue strain rate into cell-level contributions: cell-shape deformation and cell rearrangement (see *Figure 1B* and *Figure 1—figure supplement 1*). During VF invagination, cells in the lateral ectoderm are stretched along the DV axis with little rearrangement (*Figure 1—figure supplement 1A*) Subsequently, during GBE, the lateral ectoderm (i.e. the germ band) extends (*Figure 1D*) by cell rearrangements with very little cell elongation (*Figure 1C*, bottom; and *Figure 1—figure supplement 1B'*). By contrast, the dorsal tissue (which will become the amnioserosa) deforms opposite to the germ band by a combination of cell shape changes and rearrangements (*Figure 1C*, top). Overall, we find that the majority of cell rearrangements happen via T1s (*Figure 1E and E'*) with only a small number of 'rosettes' (which involve more than 4 cells transiently sharing a vertex) being formed. This prevalence of T1s is consistent with previous analyses (*Farrell et al., 2017*; *Stern et al., 2022*) and with the high coordination of rearrangements that is apparent in the lateral ectoderm (*Figure 1C*, bottom).

While the above tissue tectonics analysis reveals the cell scale 'kinetics' during GBE, it does not address the fundamental question of the underlying driving forces (*Guirao and Bellaïche, 2017*). Intercalations are associated with localized activity of the force-generating protein non-muscle myosin II (*Bertet et al., 2004*; *Zallen and Wieschaus, 2004*; *Martin, 2020*) (henceforth simply 'myosin'). However, the relative contribution of such locally generated forces vs pulling by adjacent tissues, such as the invaginating PMG remain debated. Previous studies addressing this key questions on the tissue level have come to conflicting conclusions (*Irvine and Wieschaus, 1994*; *Streichan et al., 2018*; *Collinet et al., 2015*; *Butler et al., 2009*; *Lye et al., 2015*; *Farrell et al., 2017*; *Gehrels et al., 2023*). A second key question is how myosin dynamics is controlled on a cellular level and how it is orchestrated across cells to create a coherent global morphogenetic flow. Myosin recruitment depends on the expression of key developmental genes (*Zallen and Wieschaus, 2004*; *Martin et al., 2009*; *Lecuit et al., 2011*; *Martin, 2020*), and, in addition, is subject to positive and negative mechanical feedback that depends on the stress (*Blankenship et al., 2006*; *Fernandez-Gonzalez et al., 2009*) and the rate of strain (rate of cell deformation) (*Gustafson et al., 2022*). Despite considerable understanding of the genetic and cell-biological components involved in GBE, the relative roles of the genetic pre-pattern (top-down) vs local 'self-organization' of myosin activity via mechanical feedback loops (bottom-up) are still unclear. All of these questions call for a coherent theoretical framework to interpret and reconcile existing experimental findings.

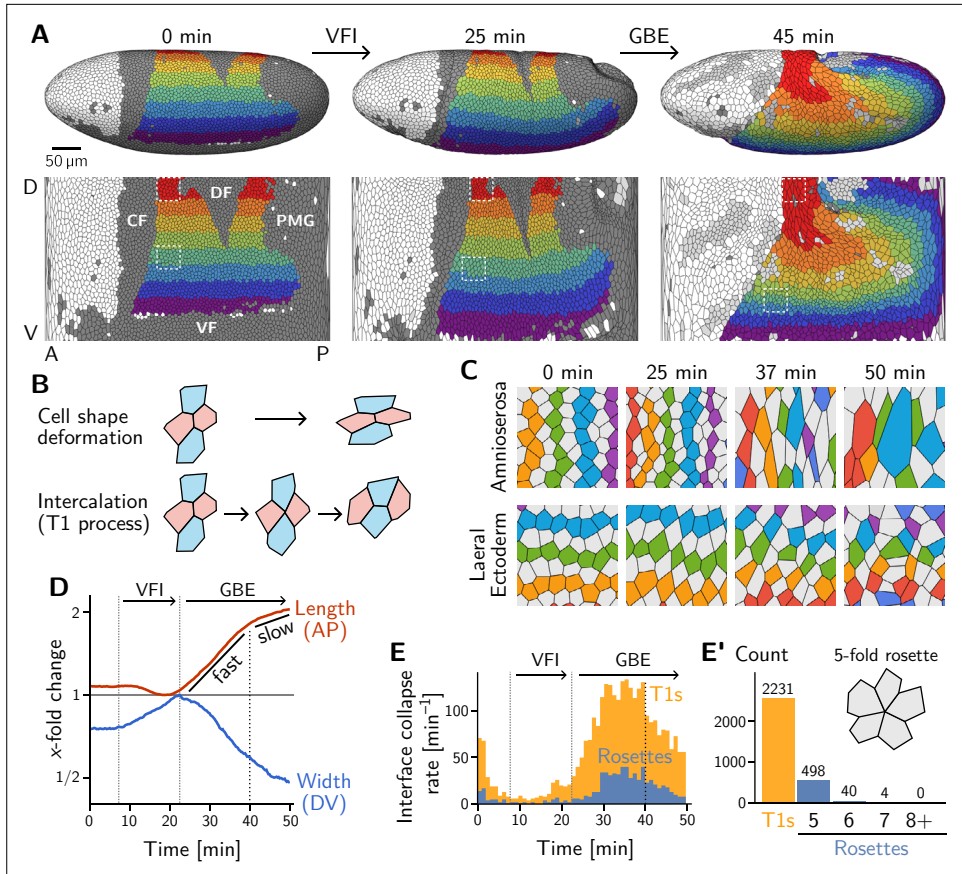

**Figure 1.** Light sheet imaging, segmentation, and tracking provide a global picture of the cell-level contributions to tissue flow. (**A**) Segmented and tracked cells on the ellipsoidal surface of the early *Drosophila* embryo in 3D (top) and projected into the plane (bottom) using a cartographic projection (*Heemskerk and Streichan, 2015*) imaging, segmentation, and tracking data from *Stern et al., 2022*. Trunk cells are colored in bands along the DV axis to illustrate the major tissue deformations during early development of *Drosophila*: ventral furrow invagination (VFI, 25 min) and germ band extension (GBE, 45 min). During GBE, the lateral ectoderm (germ band, purple to green regions) contracts along the DV axis and extends along the AP axis, causing its posterior part to move over the pole. The dorsal ectoderm (amnioserosa, red and orange regions) contracts along the AP axis and extends along the DV axis. Cells that get internalized in folds are shaded in dark gray (CF: Cephalic furrow; DF: Dorsal folds; VF: Ventral furrow; PMG: posterior midgut). Only one side of the left-right symmetric embryo is shown but both sides were analyzed throughout the manuscript. (**B**) Tissue deformation is the sum of cell shape changes (top) and cell rearrangements (bottom). The elementary cell rearrangement is a T1 transition in a quartet of cells: The interface between the red cells collapses, giving rise to a transient fourfold vertex configuration (center); the fourfold vertex then resolves to form a new interface between the blue cells. (**C**) Colored, tracked cells illustrate cell rearrangement and shape change in the amnioserosa (top) and lateral ectoderm (bottom). While amnioserosa cells show large deformations and little coordination in their rearrangement, cell intercalations in the lateral ectoderm appear highly choreographed. (ROI size $40 \times 40\,\mu\text{m}^2$). (**D**) Convergence and extension of the lateral ectoderm ($x$-fold change defined relative to the minimum length and maximum width, respectively). During VFI, the lateral ectoderm is stretched along the DV axis and slightly contracts along the AP axis. GBE has an initial fast phase before slowing down at around 40 min. (**E**) and E' Rate of interface collapses serves as a measure for the cell intercalation rate. During VFI, there are few intercalations. During GBE, a majority of intercalations are T1 transitions, while rosettes – rearrangements involving more than four cells – contribute significantly less to tissue deformation (**E'**). At 40 min, there is a noticeable drop in the T1 rate, marking the transition to the slow phase of GBE. Intercalation events before $t = 12\,\text{min}$ do not contribute to tissue flow and were excluded from the subsequent analysis.

The online version of this article includes the following figure supplement(s) for figure 1:

**Figure supplement 1.** Tissue tectonics quantification.

**Figure supplement 2.** Orientation of collapsing interfaces.

**Figure supplement 3.** Cell rearrangements preserve AP stripe pattern.

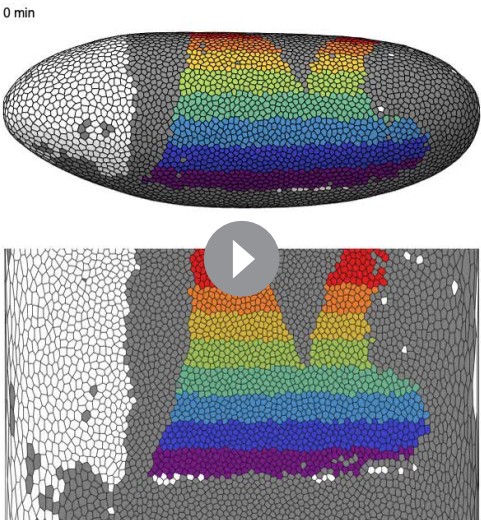

**Video 1.** *In toto* cell tracking during gastrulation. Ventral furrow invagination (gray cells on the dorsal side) is followed by convergence-extension of the germ band (lateral ectoderm cells colored purple, blue, and green). As the germ band elongates along the AP axis, the cells move over the posterior pole. The amnioserosa (orange and red cells) undergoes convergence-extension in the opposite direction of the germ band and exhibits significant cell shape elongation while cells in the germ band remain mostly isotropic in shape. Near the end of germ band extension (ca 35 min) cell divisions start. (Corresponds to *Figure 1A*; invaginating cells are colored in gray; cells in the head are colored white; cells after division events are colored in light gray).

https://elifesciences.org/articles/95521/figures#video1

Our approach is based on the assumption of force balance of stresses concentrated in the cell cortices. Intercellular adhesion effectively links the cytoskeleta of neighboring cells, so that a two-dimensional epithelial sheet constitutes a trans-cellular mechanical network put under tension by myosin motors. This internal tension is revealed by laser ablation of cell interfaces, which causes rapid recoil (*Ma et al., 2009*). Cortical force balance provides a direct link between mechanics and geometry: It allows inference of tensions from the angles at the vertices where interfaces meet (*Chiou et al., 2012*; *Ishihara et al., 2013*; *Noll et al., 2020*), which we utilize to identify from images the stereotyped local geometry and tension dynamics associated with internally driven (active) and externally driven (passive) cell rearrangements (T1s).

The capacity of epithelial tissue to support tension makes it markedly different from passive fluids (*Noll et al., 2017*). How one can reconcile the solid-like capacity of an epithelial monolayer to support tension (as well as shear stress) with its ability to change shape and rearrange internally like a fluid is intimately linked to the fundamental question of internal vs external driving *Guirao and Bellaïche, 2017*. Our findings suggest that the morphogenetic flow of epithelia can be understood as adiabatic deformation of cell array geometry controlled by changes in the internal state of tension. In other words, the tissue behaves as a plastically deforming *active solid* rather than a fluid. Based on these insights, we provide evidence and a minimal model for self-organization of the internally driven cell rearrangements via a local mechanical feedback mechanism. This model reproduces the experimentally observed dynamics on the level of cell quartets and forms the basis for a predictive tissue-scale model formulated in a companion paper (*Claussen et al., 2024*). Finally, we address how cellular behaviors (shape changes and intercalations) are coordinated across the tissue to generate coherent tissue flow. We will show that coordination of active T1 events among neighboring cells involves a characteristic pattern of cortical tensions which we quantify by introducing the 'local tension configuration (LTC)' order parameter. In *Claussen et al., 2024*, we employ this order parameter to quantitatively compare tissue-scale simulations with experimental data on the cell scale. Taken together, our findings identify the dominant role of active internal tension in the lateral ectoderm in driving the embryo scale flow and suggest mechanical feedback as the mechanism for self-organization on the cell scale.

## Results

### Force balance and cell geometry in an epithelial monolayer

We begin by laying out the assumptions and concepts that underlie our framework. Epithelial tissues are under internally generated tension, which is revealed by recoil in response to laser ablation. The timescale of this recoil ($\sim 10\,\text{s}$) on the scale of cells is at least 10-fold faster than the timescale of local tissue flow (*Bambardekar et al., 2015*; *Munjal et al., 2015*) so that the tissue can be regarded as being in approximate mechanical equilibrium. This suggests that the apparent tissue flow can be understood in terms of adiabatic remodeling of a quasistatic force balance network (*Noll et al., 2020*).

This view contrasts with regular fluid flow, where externally or internally generated forces are balanced by viscosity or substrate friction.

We further assume that mechanical stress in the epithelium is primarily carried by the adherens-junctional cytoskeleton, which resides on cell interfaces. This is supported by the observation that cell-cell interfaces in the *Drosophila* blastoderm are mostly straight. Cells are attached to their neighbors via adherens junctions that are linked to the junctional cytoskeleton in each cell (*Figure 2A*, *Lecuit and Yap, 2015*). The myosin motors exert a contractile force on the actin fibers and thereby keeping the cortex under active tension ($T$), which we refer to as 'cortical tension' (*Prost et al., 2015*). Together, the adherens junctions define a tissue-wide mechanical structure, capable of (*i*) generating locally controlled internal tension and (*ii*) adaptively remodeling its architecture. The *Drosophila* blastoderm lies on top of a fluid yolk (*Doubrovinski et al., 2017*) which exerts negligible drag forces on tissue motion on the surface (*Cheikh et al., 2022*), suggesting that all forces are balanced within the epithelial layer.

The concentration of tension in interfaces tightly links force balance with the readily observable geometry of cell-cell interfaces. Force balance requires that the forces $\mathbf{T}_{ab}$ exerted by the cortical tensions sum to zero at each vertex where three (or more) interfaces meet. (Here, the indices $a, b$ label the two cells that meet at an interface; see *Figure 2A*). These force-balance constraints relate to the *relative* tension on cell interfaces at a vertex to the angles at which the interfaces meet (*Chiou et al., 2012*). This relation between the hard-to-observe cortical tensions and the readily observable local cell geometry enables tension inference methods which have been extensively validated by computational robustness checks (*Ishihara et al., 2013*), direct comparison with measured laser ablation recoils (*Kong et al., 2019*) and by correlation with local myosin abundance (*Noll et al., 2020*).

Crucially, the link between force balance and the geometry of the cell array goes beyond inference and to the very basis of our proposed mechanism of tissue flow. Force balance implies that the tension vectors meeting at a vertex sum to zero and hence forming a a closed triangle (*Noll et al., 2017*), as illustrated in *Figure 2B*. The angles of this triangle are complementary to the corresponding angles at which the interfaces meet (This becomes evident by rotating each triangle edge by $90°$. Two angles are complimentary if they sum to 180°). As adjacent vertices share an interface, the corresponding tension triangles must share an edge: the tension triangles form a triangulation, i.e., fit together to form a tiling of the plane. The triangulation is dual to the cell array: For each cell, there is a corresponding vertex in the tension triangulation (*Figure 2B'*). It reflects the fact that the force-balance conditions at neighboring vertices are not independent because they share the interface that connects them. (Dual force tilings go back to Maxwell *Clerk Maxwell, 1864* and have been applied to the statics of beam assemblies *Varignon, 1725* and granular materials *Tighe et al., 2008*). This triangulation establishes a geometric structure in tension space. Force balance requires that the angles at vertices in the physical cell array are complementary to those in the tension triangulation. This intimately links tension space and epithelial geometry in real space. Myosin-driven local changes in tension changes the shapes of tension-space triangles and hence remodel the tension triangulation. The induced changes in the geometry of the cell array drive both the local rearrangement of cells and the global tissue flow. The remainder of the Results section will provide quantitative data analysis and modeling that will work out of the above ideas and their implications.

## Cell scale analysis

To map out the relative tensions in the tissue, we perform *local* tension inference for each interface relative to its four neighbors. Geometrically in tension space, the local tension configuration is represented by two adjoined tension triangles forming a kite (see *Figure 2B*) and the inference is a simple application of the law of sines on the two triangles (see Appendix 2). *Figure 2C and C'* show snapshots from this local tension inference in the dorsal ectoderm (amnioserosa) and the lateral ectoderm (germ band), respectively. Initially, relative tensions are close to unity throughout the embryo, since the cell array is approximately a hexagonal lattice, with vertex angles close to $120°$. As gastrulation progresses, the cortical tensions change, and one starts to see characteristic differences between the dorsal ectoderm and the lateral ectoderm. In the former, interfaces that contract remain under approximately constant tension while interfaces oriented parallel to the direction of tissue stretching (i.e. along the DV axis) extend and are under increasing tension (*Figure 2C*). A very different picture emerges in the lateral ectoderm, where one observes an alternating pattern of high and low tensions

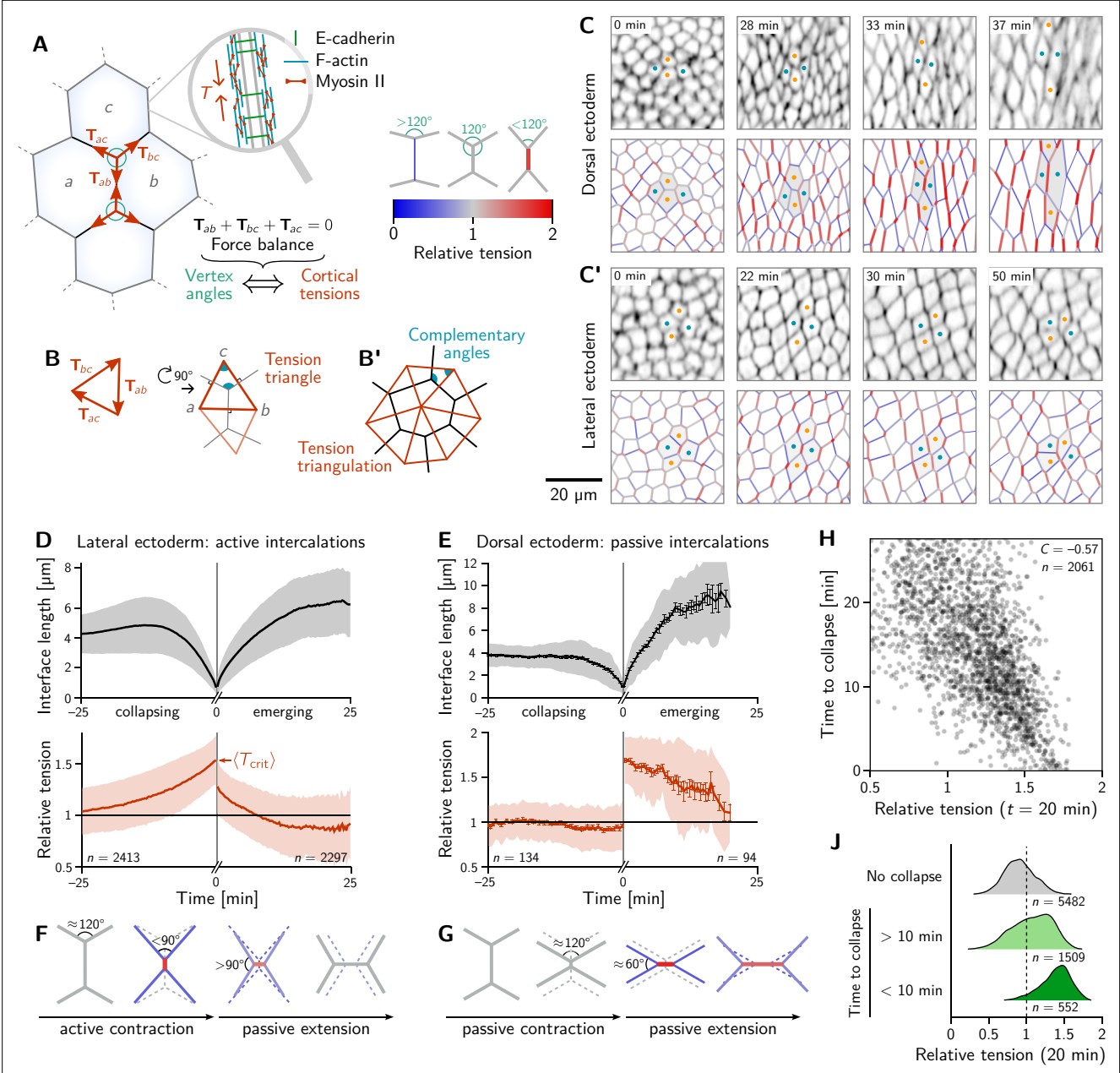

**Figure 2.** Inferred tension dynamics distinguishes active and passive T1s. (**A**) Tension in the cortex at cell-cell interfaces is actively maintained by molecular motors (inset). In force balance, the forces $\mathbf{T}_{ab}$ exerted on a vertex (red arrows) must sum to zero, thus relating angles at vertices to the relative tensions. (**B** and **B'**) The angles in the *tension triangles* formed by the force vectors (rotated by 90°) are complementary to the interface angles at the vertex. Tension triangles corresponding to adjacent vertices share an edge and therefore fit together to form a *tension triangulation* (**B'**). (**C** and **C'**) Relative tensions inferred (bottom) from cell membrane images (top) in the amnioserosa (**C**) and the lateral ectoderm (**C'**). In the lateral ectoderm, high-tension interfaces contract. The regular pattern of alternating high and low tension interfaces therefore leads to coordinated T1s (*Figure 1C*, bottom). Blue and orange dots mark an intercalating cell quartet. (**D** and **E**) While active and passive T1s show similar dynamics of the length of the inner edge (top), they are markedly different in their tension dynamics (bottom). Increasing tension on contracting interfaces provides evidence for positive tension feedback in the lateral ectoderm (**D**). Constant relative tension on contracting tensions in the dorsal ectoderm indicates passive intercalations (**E**). Tension jumps at time zero result from the relation between the angles before and after the neighbor exchange. Collapsing and emerging interfaces were tracked and analyzed separately (see Appendix 1.3). Bands and fences show SD and SEM, respectively; the SEM in (**D**) is smaller than the line thickness. (**F**) Increasing tension on an actively contracting interface causes the angles opposite of it to become increasingly acute. (**G**) Constant cortical tensions (and thus vertex angles) before the neighbor exchange are the hallmarks of passive T1s. This geometrically determines the vertex angles after the neighbor exchange, such that the emerging interface is under high tension. (**H and J**) In the lateral ectoderm, relative tension predicts the time until

*Figure 2 continued on next page*

*Figure 2 continued*

an interface collapses (**H**) and high relative tension predicts which interfaces collapse (**J**). Relative tensions were averaged from 20–21 min (over four timepoints), i.e., at the end of ventral furrow invagination (VFI) (*Figure 1E*).

The online version of this article includes the following figure supplement(s) for figure 2:

**Figure supplement 1.** Quartet analysis resolved by DV position.

**Figure supplement 2.** Relative tension does not correlate to interface collapses in the amnioserosa.

before intercalations start (22 min). The high-tension interfaces contract, leading to coordinated T1 transitions (30 min, *Figure 2C'* and *Video 2*). As GBE transitions from the fast phase to the slow phase at around 40 min, the pattern of tensions becomes more disordered. We will return to the pattern of local tension configurations below in the discussion of tissue scale dynamics.

The differences in the patterns of inferred tension in the amnioserosa compared to the lateral ectoderm suggest very distinct mechanisms for cell intercalations in these two tissue regions, matching the fact that their levels of cortical myosin are very different (high in the lateral ectoderm, low in the amnioserosa *Streichan et al., 2018*). In the following, we first focus on intercalating cell quartets to quantitatively analyze these different mechanisms. A quantitative understanding of the cell-scale dynamics will then form the basis for bridging to the tissue scale.

## Relative tension dynamics distinguishes active and passive intercalations

We identify all cell quartets that undergo neighbor exchanges (T1 processes), calculate the length and relative tension of all collapsing and emerging interfaces, and align the data to the time of the neighbor exchange (*Stern et al., 2020*). Pooling the data for each of the bands of cells colored in *Figure 1A*, we find two distinct scenarios for ventrolateral quartets and dorsal quartets (*Figure 2D and E*; for a breakdown by individual bands, see *Figure 2—figure supplement 1*).

The length dynamics of collapsing and extending interfaces in the amnioserosa and the lateral ectoderm is qualitatively similar (*Figure 2D and E*). In the lateral ectoderm, there is a slight increase in interface length preceding contraction. This transient stretching is caused by the VF invagination that precedes GBE (*Gustafson et al., 2022*).

By contrast, the tension dynamics are markedly different between the amnioserosa and the lateral ectoderm. In the lateral ectoderm, the tension on the contracting edge grows non-linearly and reaches its maximum just before the neighbor exchange. In terms of the local cell geometry, this increasing relative tension reflects the fact that the angles facing away from the interface decrease as the interface contracts (*Figure 2F*). Notably, the non-linearly increasing tension, concomitant with an accelerating rate of interface contraction, is evidence that positive tension feedback plays a role in myosin recruitment (*Blankenship et al., 2006*; *Fernandez-Gonzalez et al., 2009*; *Duda et al., 2019*) and is in excellent agreement with predictions from a recent model where such feedback is a key ingredient (*Sknepnek et al., 2023*). From the data shown in *Figure 2D* we can read off the average relative tension threshold $\langle T_{\mathrm{crit}} \rangle = 1.530 \pm 0.005$ for interface collapse. As we will see further below, this threshold can be predicted from simple geometric considerations.

The correlation between increasing tension and interface contraction during active T1s can be used to predict active T1s. Indeed, plotting the time to interface collapse against the relative tension for quartets in the lateral ectoderm shows

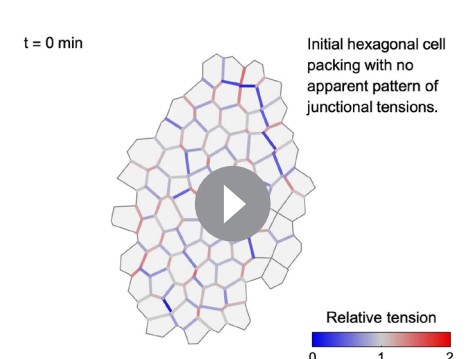

t = 0 min

Initial hexagonal cell packing with no apparent pattern of junctional tensions.

Relative tension

0    1    2

**Video 2.** Relative tension dynamics in the lateral ectoderm. Relative junctional tensions inferred from cell geometry reveal the emergence of an alternating pattern of high and low tensions that organizes cell intercalations (T1 transitions). Coherent intercalations drive convergent extension tissue flow which slows down significantly as cell scale order is lost.

https://elifesciences.org/articles/95521/figures#video2

a clear negative correlation (*Figure 2H*). Conversely, relative tensions below 1 correlate with interfaces that never collapse (*Figure 2J*).

After the neighbor exchange, the relative tension on the new interface starts at a lower value and then continues decreasing back to 1, corresponding to the vertex angles of $120°$. Because *Figure 2D and E* shows tensions on the collapsing edge for $t < 0$ and on the emerging edge for $t > 0$, there is no reason why the tension should be continuous at $t = 0$. The apparent jump in relative tension is a consequence of geometry: Because the angles facing away from the interface are $< 90°$ before the neighbor exchange, they are necessarily $> 90°$ afterward. This implies that a relative tension $> \sqrt{2}$ on the collapsing interface is necessarily followed by a tension $< \sqrt{2}$ on the new interface.

Let us now turn to T1s in the amnioserosa. Here, the relative tension in the inner edge remains almost constant near 1 prior to the neighbor exchange, i.e., the vertex angles remain close to $120°$ (see *Figure 2G*). As a consequence of this tension homeostasis on collapsing interfaces, there is no correlation between relative tension and the time until the interface collapses in the amnioserosa (see *Figure 2—figure supplement 2*). On the new interface emerging after the neighbor exchange, tension is high and remains constant for an extended period. Again, the apparent tension jump across the neighbor exchange is a consequence of geometry. Just before the neighbor exchange, the vertex angles are close to $120°$ which implies that the angles facing away from the interface are $60°$ after the neighbor exchange (see *Figure 2G*). This corresponds to a relative tension of $\sqrt{3} \approx 1.73$ on the central interface.

To understand the high tension observed on emerging interfaces, recall that tension inference only yields the *total* tension, but not on how this tension is generated in the cytoskeleton. On an extending junction, tension carried by passive crosslinkers will add to the tension generated by myosin. The passive tension rapidly relaxes as crosslinkers turn over, giving an effective viscoelastic relaxation timescale on the order of minutes (*Clément et al., 2017*). This passive tension relaxation is a crucial ingredients in the model presented below. In the amnioserosa, the high tension is sustained for a longer time because the tissue there is continually getting stretched as the germ band contracts along the DV axis. Indeed, increased tension is also found in interfaces that start out DV-oriented (*Figure 2C*).

## Minimal model based on tension feedback reproduces length- and tension-dynamics of T1s

Tension inference and the pooled analysis of T1 events have revealed the cortical tension dynamics on cell-cell interfaces during active and passive cell intercalations. The behavior of cortical tensions differs significantly between distinct spatial regions of the embryo and, at first glance, appears quite counter-intuitive. In particular, it is very different from that of springs or rubber bands. The length and tension of a springs are tied to one another by a constitutive relationship. By contrast, the length and tension of cell-cell interfaces can change independently and are actively regulated by the turn-over of junctional proteins (such as actin, myosin, and E-cadherin). Experiments where actin turnover is decreased by Cytochalasin D treatment show reduced cell intercalations supporting the point of view that the decoupling of tension regulation from a spring-like constitutive relation is important for physiological behavior.

These observations call for a new modeling approach, where tensions are not governed by constitutive relationships such as the typical area–perimeter elasticity of the vertex model (*Farhadifar et al., 2007*; *Hufnagel et al., 2007*). Instead, our model directly builds on the same assumptions that underlie the tension inference: dominant cortical tensions that are in adiabatic force balance. The dynamics reside in the changes in tension governed by mechanical feedback.

To keep the number of parameters and variables (degrees of freedom) in the model to a minimum, we consider a quartet of cells with identical shapes. This describes a representative quartet in a periodic cell array. Such a cell array is characterized by the three angles at each vertex, $\phi_i$, and the three interface lengths $\ell_i$, with $i \in \{0, 1, 2\}$ (see *Figure 3A*). The vertex angles are determined by the relative cortical tensions $T_i$ via the condition of force balance (see *Figure 2A–B*). Motivated by the nonlinearly increasing tension observed on contracting interfaces (*Figure 2D*), we equip each interface with self-reinforcing tension dynamics.

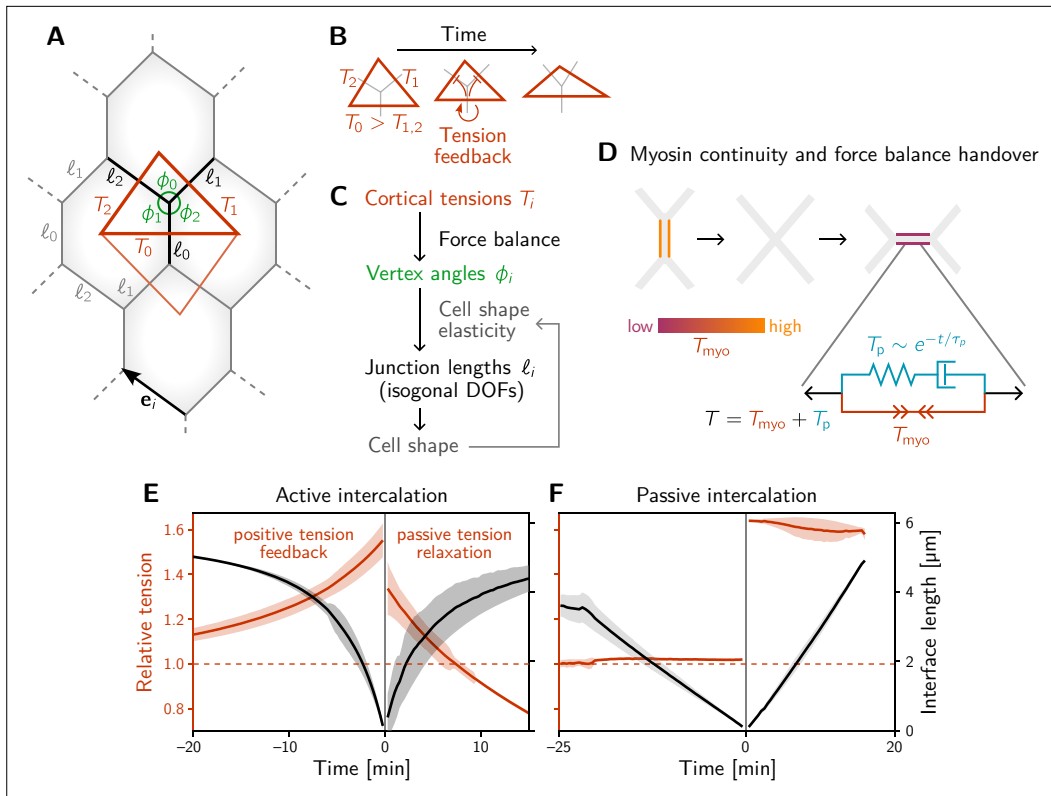

**Figure 3.** A minimal model for positive tension feedback reproduces the signatures of active T1s and creates passive T1s when feedback is turned off. (**A**) A single quartet of identical cells forms the elementary setting for modeling T1s. This geometry is characterized by two vertex angles ($\phi_0$, $\phi_1$, and $\phi_2 = 2\pi - \phi_0 - \phi_1$) and three interface lengths ($\ell_i$). The interface angles are determined by the pair of identical tension triangles corresponding to the cell quartet. To avoid boundary effects, the cell quartet and tension triangles are set up to tile the plane periodically as a regular lattice. (**B**) Positive tension feedback causes the longest edge in a tension triangle to grow at the expense of the shorter two edges, thus deforming the triangle to become increasingly obtuse (We fix the total tension scale. In real cells, the overall tension scale is set by the available myosin pool. Relative tensions change as myosin is redistributed between the cortex at different interfaces). (**C**) The tension triangle shape determines the vertex angles, $\phi_i$. To fix the interface lengths $\ell_i$, we determine the cell shape by minimizing an elastic cell-shape energy while keeping angles fixed (see Appendix 5 for details). (**D**) Two-sided architecture of junctional cortex determines the myosin level on the newly formed interface. Sketch of intercalating quartet with myosin in each cell's cortex color-coded. After a neighbor exchange, the active tension (i.e. myosin level) on the new edge is determined by a 'handover' mechanism that assumes continuity of myosin concentration at vertices within each cell. As a consequence, the active tension on the new edge right after the neighbor exchange is below the total tension that is determined by geometry. This tension imbalance causes the new edge to extend by remodeling. To capture the remodeling, we introduce a passive viscoelastic tension $T_{\text{passive}}$ due to passive cortical crosslinkers. $T_{\text{passive}}$ decays exponentially with a characteristic remodeling time $\tau_p$. Notably, no additional active ingredients (like medial myosin contractility) are required to drive the extension of the new edge. (**E and F**) The model reproduces the signatures of active and passive T1s observed in the *Drosophila* embryo. The tension feedback rate and passive relaxation rate are fitted to match the observed timescales. (Bands show the standard deviation from an ensemble of simulations with initial angles drawn from the experimental vertex angle distribution at 0 min.).

The online version of this article includes the following figure supplement(s) for figure 3:

**Figure supplement 1.** Additional quantification of single-quartet simulations.

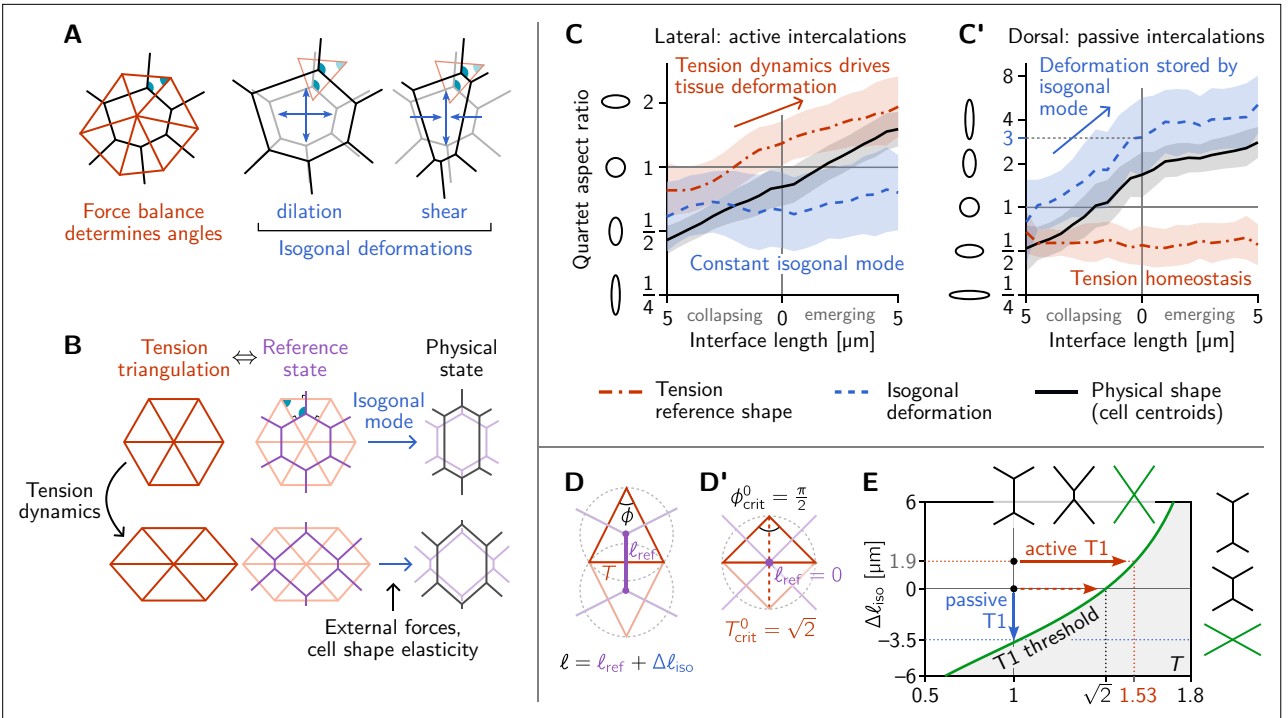

**Figure 4.** Tension–isogonal decomposition of epithelial geometry identifies active (tension-driven) and passive contributions to tissue deformation. (**A**) The angles in the tension triangulation (red) are complementary to those in the cell array (black) (left). The triangulation acts as a scaffold that leaves freedom for isogonal (angle preserving) deformations which encompass both dilation (center) and shear (right). (**B**) Deformations of the physical cell array (black, right) can be decomposed into deformations of the tension triangulation (red, left) and isogonal deformations (blue). The former reflect the dynamics of cortical tensions while the latter reflect the effect of external forces and cell shape elasticity. A reference cell array (purple, e.g. a Voronoi tessellation) constructed from the tension triangulation serves as an intermediate relative to which the isogonal deformations are defined. (**C and C'**) Quartet shape (aspect ratio) and stretch ratio of the isogonal deformation plotted against the length of the quartet's inner interface, which serves as a pseudo-time parametrization of the T1 process. An aspect ratio of 1 indicates an isotropic quartet shape and no isogonal deformation, respectively. Active T1s (left), are driven by a deformation of the tension triangulation while the isogonal mode remains constant. Passive T1s (right), are driven by isogonal deformations while the shape of the tension triangulation remains constant. Bands indicate SD; SEM is smaller than the line width. (**D**) A symmetric pair of tension triangles is characterized by a single angle $\phi$. The cell quartet's central interface in the Voronoi reference configuration (purple) connects the centers of the triangles' circumcircles (dashed gray circles). The general case of asymmetric triangles, characterized by two internal angles, is discussed in the companion paper (**Claussen et al., 2024**). (**D'**) The circumcircles coincide when $\phi = \phi_{\text{crit}}^0 = \pi/2$. In this case, the two isosceles tension triangles form a square such that we can read off the critical tension from the diagonal length $\sqrt{2}$. (**E**) T1 threshold for symmetric cell quartets in the $T$-$\Delta\ell_{\text{iso}}$ plane found by solving **Equation 3** with $\ell = 0$ for a symmetric quartet as illustrated in (**D**). The threshold can be reached by isogonal contraction under constant relative tension (blue arrow) or by active contraction under increasing relative tension (red arrow).

The online version of this article includes the following figure supplement(s) for figure 4:

**Figure supplement 1.** Additional quantification of cell quartet dynamics using the tension-isogonal decomposition.

$$\tau_{\text{T}} \partial_t T_i = T_i^n - \frac{1}{3} \sum_k T_k^n, \tag{1}$$

which models tension-induced myosin recruitment (**Blankenship et al., 2006**) (with exponent $n = 4$ in our simulation). The dynamics conserve the total active tension $\sum_i T_i$, corresponding to a finite myosin pool in each cell. (Other choices for fixing the overall scale of tension, e.g., via the triangle area, as well as other values of $n$ do not change the results qualitatively; see **Claussen et al., 2024**). The timescale of tension dynamics $\tau_{\text{T}}$ is fitted to the experimental data. Positive feedback drives the relative tension on the quartet's central interface towards the tension threshold discussed below (see **Figure 4E**) and thus drives active T1s. The interface that collapses is the one that starts with the highest initial tension, which we label with $i = 0$ by convention, such that $T_0 > T_{1,2}$.

The tension dynamics described above determine the angles at vertices via adiabatic force balance but leave the interface lengths $\ell$ as independent degrees of freedom. Further below, we will identify

these degrees of freedom as the so-called *isogonal* (angle preserving) modes, which are an immediate consequence of having cortical tensions governed by feedback instead of a constitutive relation (*Noll et al., 2017*). To find the lengths $\ell_i$, we need to account for subdominant mechanical contributions from internal structures of the cell, such as medial myosin contractility (*Collinet et al., 2015*; *Vanderleest et al., 2018*) the nucleus (*Kim et al., 2024*) and microtubules (*Ramms et al., 2013*; *Charrier et al., 2018*; *Pensalfini et al., 2023*; *Singh et al., 2024*). We account for these contributions through a 'cell elastic energy' in terms of the deviation of the cell 'shape tensor' $\mathbf{S}_\mathrm{C} = \sum_i \ell_i \mathbf{e}_i \otimes \mathbf{e}_i$ from a target shape tensor $\mathbf{S}_0$.

$$E_\mathcal{C} = \lambda \left[\mathrm{Tr}(S_\mathcal{C} - S_0)\right]^2 + \mu \, \mathrm{Tr}[(S_\mathcal{C} - S_0)^2] \qquad (2)$$

Here, $\mathbf{e}_i$ are the unit vectors pointing along the interfaces with lengths $|\mathbf{e}_i| = \ell_i$ (see *Figure 3A*) and $\otimes$ denotes the tensor product. We choose this shape tensor because it is linear in interface length, meaning that it does not change if an interface is subdivided by inserting an additional vertex.

Importantly, we assume the scale of $E_\mathcal{C}$ to be much smaller than the elastic energy due to cortical tensions. Because of this separation of scales, the angles at vertices are fixed by tensions, and the relaxation of the cell elastic energy only affects interface lengths. The coefficients $\lambda$ and $\mu$ control the cell's resistance to isotropic compression/dilation and shear deformations, respectively (see Appendix 5 for details). Notably, this elastic energy does not engender an 'energy barrier' for T1 transitions as is found in vertex models with area–perimeter elasticity (*Bi et al., 2015*) (see also *Figure 3—figure supplement 1D*). Therefore, our simulations do not require stochastic fluctuations to drive T1s.

The model described thus far captures the dynamics up to the neighbor exchange (time $< 0$ in *Figure 3E, F*) and qualitatively reproduces the dynamics of interface length and tension observed in the experimental data (*Figure 2D, E*). Interface contraction during an active T1 is driven by active remodeling of the force balance geometry: Myosin recruitment increases the cortical tension and thus drives it out of force balance with the external tension from adjacent interfaces. As a result, the interfaces remodel, changing the angles at vertices until force balance is reestablished. Positive tension feedback continually causes further myosin recruitment, which in turn drives further contraction until the interface has fully collapsed.

How does the subsequent resolution and interface elongation work? On the emerging interface, there is typically little myosin (*Bertet et al., 2004*; *Rauzi et al., 2008*). Therefore, active contractility cannot balance the external tension exerted on the interface by its neighbors: The new interface is thrown out of active force balance. This force imbalance naturally leads to the extension of the new edge. As the interface extends, passive tension-bearing elements of the cytoskeleton, such as cross-linkers, get loaded until force balance is reestablished. To account for this, we split the total cortical tension $T$ into the myosin-borne, active tension $T_\mathrm{myo}$ and the passive tension $T_\mathrm{p}$. Cortical remodeling due to turnover of crosslinkers will gradually relax this passive tension. We account for this by an exponential decay $T_\mathrm{p} \propto e^{-t/\tau_\mathrm{p}}$ with a characteristic remodeling rate $\tau_\mathrm{p}$. This passive tension effectively represents Maxwell-type viscoelasticity of the extending interface, illustrated by the spring and dashpot in *Figure 3D*. Such visco-elastic behavior on a scale of minutes has been reported in experimental data from the *Drosophila* embryo (*Clément et al., 2017*).

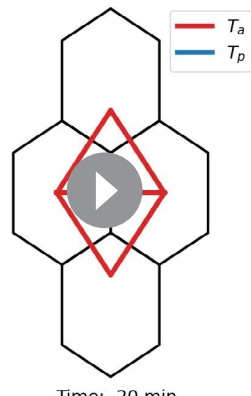

Time: -20 min

**Video 3.** Simulation of a single intercalating cell quartet. Simulation of an intercalating cell quartet driven by positive tension feedback and myosin handover mechanism, corresponding to *Figure 3E*. Out of the ensemble from *Figure 3E*, the movie and shows a simulation for symmetric initial tensions (i.e. equal initial tensions on the two non-collapsing interfaces $T_1 = T_2$). After the edge flip, the blue parts of the inner tension triangulation edge indicate the passive tension on the newly formed interface. The passive tension rapidly relaxes while the active tension grows due to positive tension feedback.
https://elifesciences.org/articles/95521/figures#video3

Completing the model requires understanding what sets the initial motor protein level on an emerging interface, which will set the initial condition for the tension dynamics on that interface. We propose a myosin 'handover' mechanism based on the assumption that the myosin level along the cortex within each cell changes continuously along the interfaces and across vertices (*Kale et al., 2018*; see *Figure 3D*). (Force balance requires that the total tension, the sum of the tensions in the two abutting cortices, is uniform along an interface. However, the individual tensions on either side can be non-uniform, as the resulting traction forces are exchanged via E-cadherin molecules linking the cells along the interface *Chiou et al., 2012*). Importantly, this 'handover' model predicts that the myosin level (*not* the total tension) on the newly formed interface is always lower than that on the collapsing interface, in agreement with experimental observations (*Bertet et al., 2004*; *Rauzi et al., 2008*).

Simulating this model, we find good qualitative agreement between the single-quartet model and experimental data, both for the tension dynamics (compare *Figure 3E and F* and *Figure 2D and E*) and for the tension–isogonal decomposition which we will introduce below (compare *Figure 3— figure supplement 1A, A'* and *Figure 4C and C'*). *Video 3* shows the simulation of an active T1 transition for a symmetric quartet. In the absence of positive tension feedback, the model successfully describes passive intercalations when the displacements of cell centroids are prescribed to mimic external stresses (*Figure 3F*; see Appendix 5 for details).

Notably, in our model, there is no additional active mechanism for interface extension. Instead, interface extension is a consequence of the fundamental temporal asymmetry of T1 processes: high myosin levels on the collapsing interface versus low myosin levels on the emerging interface. No additional active ingredients are required.

Here, we considered a minimal model for an idealized regular lattice geometry. In a real tissue, there will inevitably be disorder which impacts how cells can coordinate with one another. In the companion paper (*Claussen et al., 2024*), we investigate the role of such disorder in tissue-scale simulations comprising many cells. For the remainder of this manuscript, we return to the analysis of experimental data, going from the cell scale to the tissue scale.

## A tension–isogonal decomposition quantifies active vs passive contributions to tissue deformation

The above analysis and minimal model have revealed the distinct mechanical forces that drive active and passive cell intercalations. These forces manifested themselves through their effect on the geometry of the cell array. Dominant cortical tensions constrain the angles at vertices. These angle constraints are represented by the tension triangulation. Interface lengths can change collectively while keeping those angles fixed via isogonal modes (*Noll et al., 2017*), as illustrated in *Figure 4A*. These isogonal modes, therefore, represent cell and tissue deformations that take place under *constant cortical tensions*. They can be caused, for instance, by external forces and cell-internal elasticity (medial myosin, stiffness of the nucleus, intermediate filaments). It is important to keep in mind that this association of isogonal deformations with non-junctional stresses are based on the assumption that (active) cortical stresses are the dominant source of stress in the tissue (separation of scales).

These above considerations suggest that we can use the observed geometry of the cell array to decompose cell and tissue deformations into two distinct components: (*i*) deformations of the tension triangulation reflecting changing cortical tensions, and (*ii*) isogonal deformations reflecting all other mechanical forces. We will call this a *tension–isogonal decomposition*. This decomposition only assumes the dominance of cortical tensions over other stresses in the tissue and in particular is independent of the specific form of the cell elastic energy *Equation 2* we used in the minimal model above.

The key elements of the tension–isogonal decomposition are illustrated in *Figure 4B*. Changes in cortical tension are reflected in deformations of the tension triangulation (i.e. changes in the angles at vertices). For each tension triangulation, we can construct a reference state for the polygonal cell array – compatible with the force balance constraints – from the tension triangulation using a Voronoi tessellation. Isogonal deformations then transform this reference state into the physical cell array. In practice, we do not need to explicitly construct the reference cell array. Instead, we utilize the fact that the physical cell array can be quantified by the triangulation defined by the cells' centroids (*Merkel*

*et al., 2017*). The isogonal modes are calculated as the transformations that deform the tension triangulation into the centroidal triangulation.

We first apply the tension–isogonal decomposition to intercalating quartets. For a given quartet of cells labeled $i = 1-4$, we quantify the physical shape in terms of the centroid positions $\mathbf{c}_i$ by calculating the moment of inertia tensor. The tension reference shape is given by the tension vectors $\hat{\mathbf{T}}_{ij}$, which are obtained from the interface angles using tension inference and are rotated by $90°$ relative to the interfaces (see *Figure 2B*). The ratios of eigenvalues of the moment-of-inertia tensors define the shape aspect ratios, illustrated by ellipses in *Figure 4C and C'*. The quartet's isogonal deformation tensor $\mathbf{I}_{\mathrm{quartet}}$ transforms the cell centroids into the tension vertices $\ell_0 \, \mathbf{I}_{\mathrm{quartet}}.\hat{\mathbf{T}}_{ij} = \mathbf{c}_i - \mathbf{c}_j$ (see *Figure 4—figure supplement 1A* for a labeled sketch). Given $\hat{\mathbf{T}}_{ij}$ and $\mathbf{c}_i$, the equations can be (approximately) solved for the best fitting $\mathbf{I}_{\mathrm{quartet}}$ using least squares (an exact solution is not possible in general because the equations are overdetermined; see Appendix 3). The constant global scale factor $\ell_0 \approx 4.2 \, \mu\mathrm{m}$ translates units from relative tensions (a.u.) to length ($\mu\mathrm{m}$) and can be calculated from the average interface length $\langle \ell(0) \rangle \approx 3.5 \, \mu\mathrm{m}$ at the first timepoint using the Voronoi–Delaunay duality as explained in Appendix 3.

In *Figure 4C and C'*, we plot the quartet shape aspect ratios and the eigenvalue ratio of the isogonal deformation tensor against the length of the quartet's inner edge. The latter serves as a pseudo-time parameterization of the T1 process; plots against real-time are shown in *Figure 4—figure supplement 1B, B'*. In the lateral ectoderm, tension-triangle deformation accounts for quartet shape deformation, indicating that tension dynamics drives the intercalation (*Figure 4C*). This confirms our conclusion from the relative tension analysis (*Figure 2D*). The isogonal strain is approximately constant and, in fact, slightly counteracts the deformation. This is a consequence of the ventral furrow invagination which pulls on the lateral ectoderm cells and passively stretches them along the DV axis. A very different picture emerges in the passively deforming amnioserosa (*Figure 4C'*). Here, the tension mode is constant—indicating tension homeostasis—while the cell quartet deformation is entirely stored in the isogonal mode. While the behaviors observed in the lateral ectoderm and the amnioserosa represent the extreme cases, being purely tension-driven (active) in the former and purely isogonal (passive) in the latter, there is a continuous spectrum in between. Intermediate scenarios where both cortical tensions and external stresses contribute to tissue deformation are conceivable and can be detected and analyzed using the tension–isogonal decomposition.

Taken together, the tension–isogonal decomposition quantitatively distinguishes between active and passive intercalations. Tensions are controlled locally, while the isogonal modes accommodate external, non-local forces, such as pulling by adjacent tissue. We associate isogonal deformations with passive deformations because active stresses in the lateral ectoderm are generated at interfaces. In systems where active stresses are generated in the cell's interior, e.g., due to nuclear migration (*Bocanegra-Moreno et al., 2023*) or due to intracellular actin cables (*Priess and Hirsh, 1986*), the isogonal mode is actively controlled. Another example of this is apical constriction during VF invagination which is driven by medial myosin pulses that drive isogonal cell contractions (*Noll et al., 2017*).

## Tension space geometry sets the tension threshold for T1 events

A puzzle that has remained open so far is the threshold value of relative tension where a neighbor exchange takes place. In the lateral ectoderm where the isogonal strain is approximately constant (*Figure 4C*) the neighbor exchange takes place at a critical relative tension of $1.530 \pm 0.005$ on average (*Figure 2D*). By contrast, in the amnioserosa, where T1s take place under constant tension (*Figure 2E*), there is an average critical aspect ratio $\sim 3$ of the isogonal deformation at the point of neighbor exchange (see *Figure 4C'*).

We now explain these numbers in terms of the geometric tension–isogonal decomposition introduced in the previous section. We start by decomposing the length $\ell$ of a cell quartet's central interface into a reference length $\ell_{\mathrm{ref}}(\{T_i\})$, determined purely in terms of the local relative tensions, and an isogonal length $\Delta \ell_{\mathrm{iso}}$ depending on the quartet's isogonal strain:

$$\ell = \ell_{\mathrm{ref}}(\{T_i\}) + \Delta \ell_{\mathrm{iso}}. \tag{3}$$

A neighbor exchange takes place when $\ell$ shrinks to zero. Thus, setting $\ell = 0$ in *Equation 3* defines the threshold values of the local tensions $\{T_i\}$ and the isogonal length $\Delta \ell_{\mathrm{iso}}$ where T1s take place (see

green line in *Figure 4E*). It is immediately apparent that the interface collapse can be driven by the isogonal mode, by changes in tensions, or by a mixture of both.

Let us start with the passive T1s observed near the dorsal pole. In this case, the relative tensions are homeostatic (*Figure 2E*), so $\ell_{ref} \approx \langle \ell(0) \rangle \approx 3.5\,\mu m$ remains constant. The neighbor exchange, therefore, happens when $\Delta \ell_{iso} = -\langle \ell(0) \rangle$ (see blue arrow in *Figure 4E*). By a geometric construction (*Figure 4—figure supplement 1D*) one finds that the corresponding isogonal deformation has an aspect ratio of 3, in good agreement with observations (*Figure 4C'*).

To find the tension threshold for active T1s, we calculate $\ell_{ref}$ using the Voronoi construction based on the vertices of the tension triangulation (see Appendix 3 for details). The length of a quartet's central interface in the Voronoi reference state is determined by the pair of tension triangles corresponding to the cell quartet as illustrated in *Figure 4D*. For simplicity, we consider the case of two identical, isosceles tension triangles which are fully characterized by the relative tension $T = 2\sin(\phi/2)$ on the central interface, with $\phi$ the angle in the tension triangle. (The general case of asymmetric tensions is discussed in the companion paper *Claussen et al., 2024*, where we show that the T1 threshold is shifted towards stronger tension anisotropy with increasing asymmetry of the tension configurations). In the symmetric case, the Voronoi construction for the reference length yields $\ell_{ref} = \ell_0 T \cot(\phi)$. In the absence of isogonal deformations ($\Delta \ell_{iso} = 0$), the interface length vanishes when $\phi = \pi/2$, implying the critical relative tension $T_{crit}^0 = T_{crit}(\Delta \ell_{iso} = 0) = \sqrt{2} \approx 1.41$. One can understand the value $\sqrt{2}$ for the critical relative tension directly from the circumcircle construction of the Voronoi tessellation as shown in *Figure 4D'*. $\ell_{ref}$ vanishes when the two adjacent tension triangles share the same circumcircle. In the case of isosceles triangles, this implies that they form a square such that we can read off the critical relative tension from the length of the diagonal.

In general, the condition $\ell_{ref}(T) = -\Delta \ell_{iso}$ for the neighbor exchange defines the 'T1 threshold' in the $\Delta \ell_{iso}$-$T$ plane (see *Figure 4E*). This diagram quantifies how active and passive forces interact to drive cell intercalations. Active tension dynamics and passive isogonal strain appear as orthogonal ways to drive T1s, as illustrated by the red and blue arrows.

In the following section, we will generalize the tension–isogonal decomposition to the tissue scale. From this analysis, one can then estimate a local average value for $\Delta \ell_{iso}$ and then read off the T1 threshold from *Figure 4E*. This will allow us to explain the value $T_{crit} \approx 1.53$.

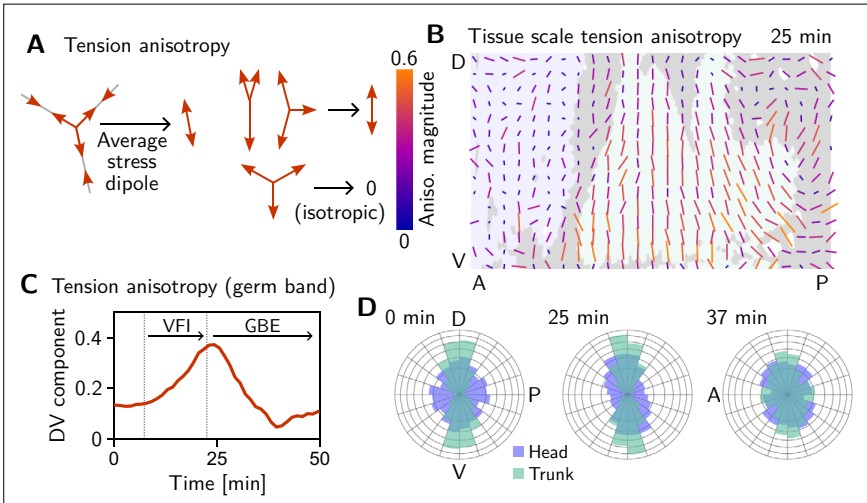

**Figure 5.** Tissue scale tension anisotropy matches orientation of convergent extension flow. (**A**) Local anisotropy of tension (double-ended arrow) at a single tri-cellular vertex. (**B**) Tension anisotropy at the end of VF invagination/ onset of GBE (25 min) locally averaged on a grid with $20\,\mu m$ spacing. Line segments indicate the local orientation and magnitude (length and color of the line segments) of tension anisotropy. (**C**) Mean DV component of locally averaged anisotropic tension in the trunk (green region in B) (DV component measured along a fixed axis orthogonal to the long axis of the embryo; SE is smaller than the line width). (**D**) Significant DV alignment of the tension orientation in the trunk precedes any tissue flow, while the tension in the head shows no orientation bias (0 min). The DV alignment of tension slightly increases during VF invagination (25 min) and decreases during GBE (37 min).

## Tissue scale

Our analysis so far has focused on the mechanism of individual T1 transitions in cell quartets. To bridge the gap from cell quartets to tissue-scale convergent extension flow, we need to address three key questions: (*i*) How are active T1s oriented? (*ii*) Which regions of the tissue deform actively due to internally generated local tension and which ones yield passively to stresses created by adjacent regions? (*iii*) How are active T1s coordinated across cells, so that different interfaces do not 'attempt' to execute incompatible T1s? In the following, we will address these three questions by building on the tools (tension inference and tension–isogonal decomposition) we employed above to analyze intercalating cell quartets.

## Initial anisotropy of tension matches orientation of flow

Convergent extension flow of tissue requires that the T1 transitions are oriented. If cortical tensions are regulated by positive feedback, the tissue-scale tension anisotropy sets the orientation of the interfaces which will collapse, and hence, the direction of tissue flow. Therefore, tissue-scale anisotropy of active tension is central to driving and orienting convergent extension flow (*Rauzi et al., 2008*; *Etournay et al., 2015*; *Lau et al., 2015*; *Streichan et al., 2018*).

We assess tension anisotropy by locally averaging the anisotropy of inferred tensions $\mathbf{T}_{ij}$ at a given individual vertices as illustrated in *Figure 5A* (see Appendix 2 for details). The inferred tension anisotropy (*Figure 5B*) and its time course (*Figure 5C*) are consistent with previously published experimental observations: During VF invagination, DV-oriented interfaces in the lateral ectoderm are stretched causing myosin recruitment (*Gustafson et al., 2022*), increasing tension anisotropy. Remarkably, we find strong DV alignment of tension anisotropy in the trunk already before VF invagination (*Figure 5D*, 0 min). This supports our hypothesis that tension anisotropy is set up in the initial condition. As GBE progresses, the DV alignment of tension anisotropy decreases (*Figure 5C and D*, 37 min). Numerical simulations presented in the companion paper (*Claussen et al., 2024*) reproduce this loss of global tension anisotropy and show that it is responsible for the slowdown of GBE after a twofold extension.

## Isogonal strain identifies regions of passive tissue deformation

We began our investigation with a 'tissue tectonics' analysis (*Blanchard et al., 2009*) that identifies the contributions of cell intercalations and cell shape changes to tissue deformations. This kinematic quantification in itself is not informative of the mechanical forces driving epithelial dynamics. To quantify the relative contributions of active (local) vs passive (non-local) forces acting on intercalating cell quartets we introduced the tension–isogonal decomposition (*Figure 4B*). Like a tissue tectonics analysis, this decomposition can be performed on the scale of entire tissue regions without the need to track cells and identify T1 events. Specifically, we exploit the fact that the isogonal strain tensor can be calculated for each individual triplet of cells that meet at a vertex (see *Figure 6A* and Appendix 3 for details). Locally averaging over nearby vertices then yields the tissue-scale isogonal deformation (see *Figure 6B and B'*). At the end of VF invagination (25 min), significant isogonal strain has built up adjacent to the VF indicating that the tissue there is passively stretched by the invaginating VF (see the purple shaded region in *Figure 6B*). The lateral ectoderm further dorsal also accumulates some isogonal extension along the DV axis (green shaded region, see time traces in *Figure 6C*). Specifically, from the DV-DV component of the isogonal strain tensor, we can estimate the average isogonal contribution to the length of DV-oriented interfaces $\langle \Delta \ell_{\mathrm{iso}} \rangle \approx 1.9 \, \mu\mathrm{m}$ in the lateral ectoderm. Using this value, one can read off the predicted relative tension threshold $T_{\mathrm{crit}} \approx 1.53$ in from the green curve in *Figure 4*. This value is in excellent agreement with the value found by tension inference (*Figure 2D*), thus validating the model for active T1s. An immediate prediction from this model is that abolishing VF invagination will eliminate isogonal stretching of the lateral ectoderm and thereby shift the T1 threshold to $T_{\mathrm{crit}} = \sqrt{2}$. To test this prediction, we analyzed a light-sheet recording of a *snail* mutant from the *Drosophila* morphodynamic atlas (*Mitchell et al., 2022*). This mutant lacks a VF as illustrated in *Figure 7A* (*Gheisari et al., 2020*). As expected, we find almost no isogonal strain in the lateral ectoderm of the *snail* mutant (see *Figure 7—figure supplement 1*). Analyzing the dynamics of relative tension in intercalating quartets, we find $T_{\mathrm{crit}} \approx \sqrt{2}$ (see *Figure 7B*), confirming that tension–isogonal decomposition faithfully captures the interplay of internally generated stresses and external forces during T1 transitions.

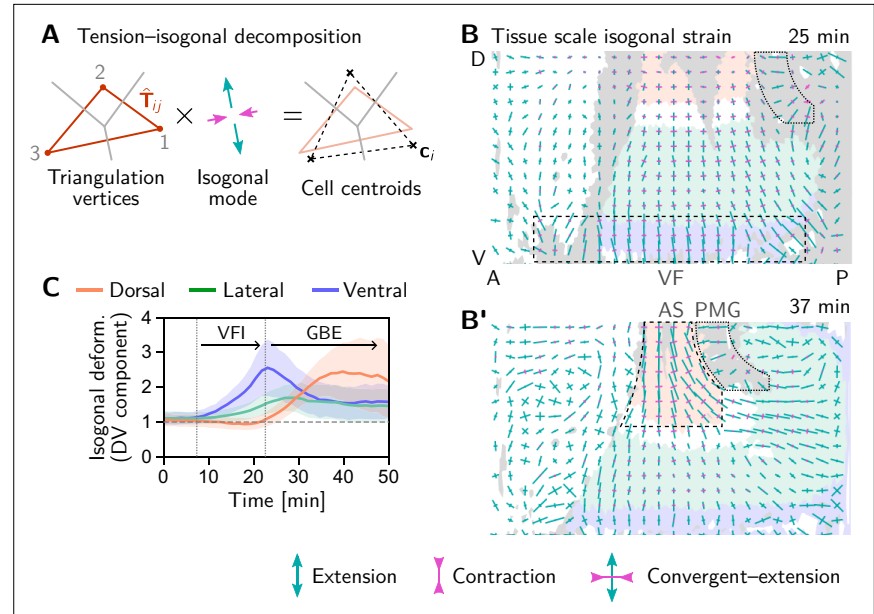

**Figure 6.** Tissue scale quantification of isogonal strain identifies regions of passive tissue deformation. (**A**) Tension-isogonal decomposition at a single tri-cellular vertex. The isogonal strain tensor (illustrated by blue arrows) transforms the tension triangle (solid red lines) into the centroidal triangle (dashed black lines). (**B and B'**) Isogonal strain at the end of VF invagination (25 min, **B**) and during GBE (37 min, **B'**) averaged over vertices in a grid with $20\,\mu m$ spacing. High isogonal strain in the tissue adjacent to the VF at 25 min (dashed black rectangle) and in the amnioserosa (AS, dashed black outline) at 37 min indicate passive tissue deformations in these regions. High isogonal strain is also found at the front of the invaginating posterior midgut (PMG, dotted outline). Crosses indicate the principal axes of isogonal strain. Bar lengths indicate the magnitude of strain (green: extensile, magenta: contractile). Colored tissue regions are quantified in (**C**). (**C**) Time traces of the DV component of isogonal strain. The isogonal (i.e. passive) stretching of the tissue adjacent to the VF (purple) is transient. The lateral ectoderm as a whole (green and blue) is stretched weakly, but persistently. The amnioserosa (red) is strongly stretched as the lateral ectoderm contracts along the DV axis during GBE (DV component is defined with respect to the local co-rotating frame, see SI; Shaded bands show one SD; SEM is comparable to the line width). *Figure 6—figure supplement 1*. Illustration and additional quantification of tissue-scale tension-isogonal decomposition.

The online version of this article includes the following figure supplement(s) for figure 6:

**Figure supplement 1.** Illustration and additional quantification of tissue-scale tension-isogonal decomposition.

During GBE, the isogonal strain near the ventral furrow relaxes back to nearly isotropic, suggesting that the passive deformation stored in the isogonal mode is elastic rather than plastic (*Figure 6B' and C*). At the same time, the actively contracting lateral ectoderm stretches the dorsal tissue (amnioserosa) along the DV axis. This passive deformation of the amnioserosa is consistent with the transition of cells from columnar to squamous (*Stern et al., 2022*) and low myosin density in the amnioserosa (*Streichan et al., 2018*). There is also significant isogonal strain just anterior of the PMG (highlighted by the dashed outline in *Figure 6B'*), as well as on the lateral side of the PMG, suggesting that the convergent extension tissue flow near the dorsal pole (*Streichan et al., 2018*) exerts a pulling force on the PMG. In this picture, the tissue on the dorsal pole is stretched along the DV axis by active convergent extension in the germ band and contracts along the AP axis to conserve total cell area, leading to an anterior-ward pulling force on the PMG. Quantitative re-analysis of previously published cauterization experiments (*Collinet et al., 2015*) provides independent evidence of such a pulling force (see Appendix 6).

Taken together, isogonal strain identifies regions where passive tissue deformation is caused by mechanical coupling across the tissue. Importantly, the absence of isogonal strain along the AP axis in the germ band provides strong evidence that the main driver of GBE is internally generated stress, rather than an external pulling force from the PMG. This conclusion from cell-scale analysis is in line

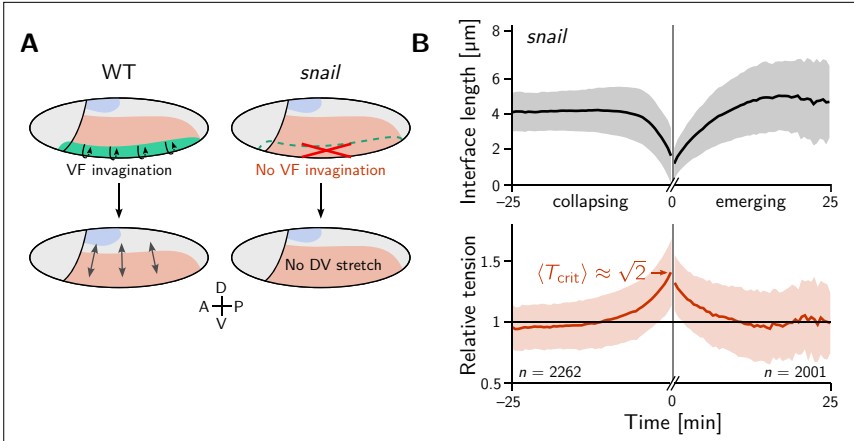

**Figure 7.** The relative tension threshold for T1s is shifted in a *snail* mutant. (**A**) In a wild-type (WT) embryo (left), the invaginating ventral furrow (green) stretches the lateral ectoderm (red) along the DV axis prior to germ band extension (GBE). In a *snail* mutant (right), the ventral furrow is abolished, such that no DV-stretch of the lateral ectoderm occurs. (**B**) In a snail mutant, the T1 threshold of active T1s in the lateral ectoderm is shifted to the value $\sqrt{2}$ predicted by our model in the absence of isogonal deformation (*Figure 4E*). *Figure 7—figure supplement 1*. Quantification of isogonal strain in a *snail* mutant.

The online version of this article includes the following figure supplement(s) for figure 7:

**Figure supplement 1.** Quantification of isogonal strain in a *snail* mutant.

with quantitative tissue-scale analysis of previously published experimental data which we address in the Discussion.

## A geometric order parameter quantifies local tension configurations

Tissue-scale tension anisotropy is not enough to drive coherent 'parallel' T1s. Such coordination requires an alternating pattern of high and low tensions as seen in *Figure 8A*, center (*Figure 2C'*). In contrast, when several connected interfaces are under high tension, they form a tension cable, contraction of which leads to rosette formation (*Blankenship et al., 2006*; *Harding et al., 2014*). Tension patterns do not appear globally with perfect regularity, but rather as local motifs that can be distinguished based on the tension configuration at individual vertices. These configurations correspond to characteristic triangle shapes in the tension triangulation (see *Figure 8B*): The elementary motif of the alternating tension pattern is a 'tension bridge' where a high-tension interface is surrounded by low-tension interfaces (see *Figure 8A*, right), corresponding to an obtuse triangle (*Figure 8B*, top right). In contrast, an acute tension triangle (*Figure 8F*, bottom right) corresponds to the local motif of a tension cable, where neighboring interfaces along the cable are under high tension while those transverse to the cable are under low tension (see *Figure 8A*, left). We can, therefore, use the tension triangle shape to define a LTC parameter. The LTC-parameter space (i.e. the space of triangle shapes), shown in *Figure 8B*, is spanned by two axes, measuring how elongated and how acute or obtuse the tension triangle is (see *Claussen et al., 2024*, *Figure 8—figure supplement 1* and Appendix 4 for details). These axes correspond to the magnitude of local tension anisotropy and the cable vs bridge character of the local tension configuration, respectively.

## LTC order choreographs T1s

The geometric condition for neighbor exchanges discussed above defines a threshold in the LTC-parameter space (dashed line in *Figure 8B*, see *Claussen et al., 2024* for details). When the shapes of a pair of adjacent tension triangles cross this threshold, the interface corresponding to the triangles' shared edge collapses, giving rise to a T1 transition. From the position of the T1 threshold in the LTC-parameter space, it is immediately evident that for tension cables (acute tension triangles), the tension anisotropy required to cross the T1 threshold is higher compared to tension bridges. Tension cables are, therefore, less efficient at driving T1s: they require stronger anisotropy of tension to cause an interface collapse.

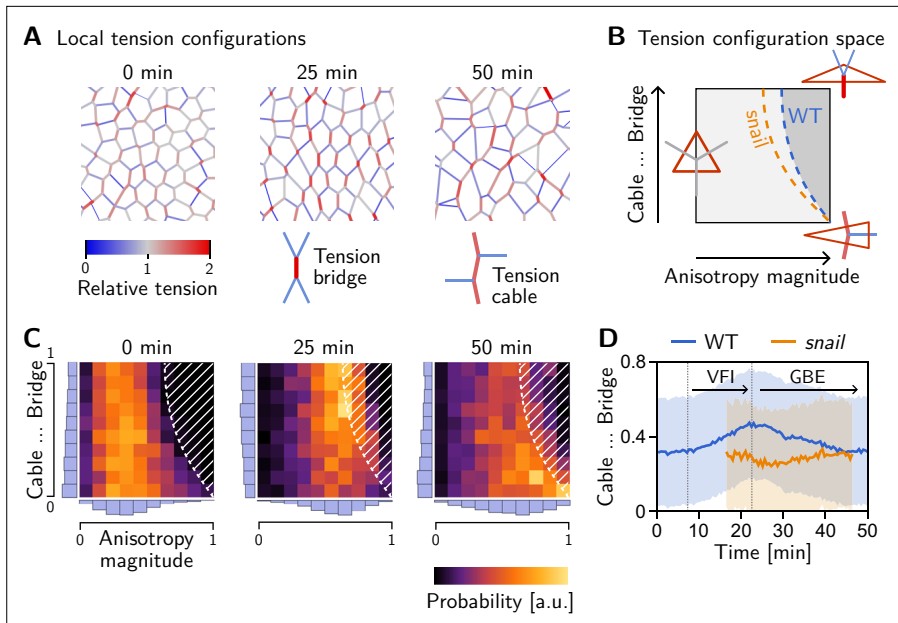

**Figure 8.** Emergence and loss of order in local tension configurations. (**A**) Distinct configurations of tension are found on the cell scale: 'Tension bridges,' characterized by a high-tension interface connected to four low-tension interfaces, are the local motif of an alternating pattern of high and low tensions. This alternating pattern gives rise to coordinated T1s as the high-tension interfaces collapse, driven by positive tension feedback. By contrast, tension cables, characterized by multiple adjacent high-tension interfaces, cause frustrated or incoherent T1s which manifest as rosettes. (**B**) Space of local tension configurations at a single vertex (quantified by the tension triangle shape, see *Claussen et al., 2024* and Appendix 4 for details). The dashed lines indicate the 'T1 thresholds' in the absence of isogonal strain (orange line, '*snail*') and for the average isogonal strain in the wild-type (WT) embryo's germ band (blue line). This threshold is at a lower anisotropy magnitude for tension bridges than for tension cables, indicating that the former are more efficient at driving active intercalations. (**C**) Distribution of tension configurations defines an order parameter that quantifies the relative abundance of tension cables and bridges in the lateral ectoderm. Arrows highlight the increasing fraction of tension bridges before germ band extension (GBE) (0–25 min) and its decrease during GBE (25–50 min). (**D**) Median of the bridge-vs-cable parameter in the lateral ectoderm of the WT embryo (blue line) and the *snail* embryo (orange line). (The initiation of cephalic furrow invagination was used for temporal alignment. Shading shows SD; SEM is smaller than the line width).

*Figure 8—figure supplement 1*. Quantification of local tension configurations in terms of tension triangle shape.
*Figure 8—figure supplement 2*. Shape statistics of a random Delaunay triangulation.

The online version of this article includes the following figure supplement(s) for figure 8:

**Figure supplement 1.** Quantification of local tension configurations (LTC) in terms of tension triangle shape.

**Figure supplement 2.** Shape statistics of a random Delaunay triangulation.

---

Two-dimensional histograms of the LTC-parameter distribution in the lateral ectoderm show that the fraction tension of bridges increases before the onset of GBE (see *Figure 8C*). The bridge fraction reaches its maximum around 25 min, just at the onset of GBE (see *Figure 8D*). The corresponding alternating pattern of tensions is clearly visible in *Figure 8A* (center). This suggests that the pattern of tensions on the (sub-)cellular scale is biologically relevant to choreographing parallel T1s. As tension anisotropy increases (*Figure 5C*), the local tension configurations eventually hit the T1 threshold where a cell neighbor exchange occurs, causing a reconfiguration of local tensions. Therefore, tension configurations beyond the T1 threshold are strongly suppressed (hatched region in *Figure 8C*). We observe that as cell intercalations take place in the lateral ectoderm, the fraction of tension bridges decreases and the fraction of tension cables increases (see *Figure 8C*, 50 min; and *Figure 8D*). In fact, the distribution of tension triangle shapes appears to approach that of a random Delaunay triangulation (see *Figure 8—figure supplement 2*), a completely disordered distribution. This suggests that the triangle edge flips can be statistically understood as a random 'mixing' process that generically causes a loss of LTC order and global alignment of tension anisotropy. This result, as well as the

findings of the tissue-scale analysis more generally, are reproduced by tissue-scale simulations of our minimal model, as shown in the companion paper *Claussen et al., 2024*.

Recall that in a *snail* mutant, the lack of DV-stretching of the lateral ectoderm implies that T1s occur at a reduced tension anisotropy compared to the wild type. *Figure 8B* shows that the shifted T1 threshold affects tension bridges more than tension cables. As a result, we expect that tension-bridge configurations undergo T1s more rapidly in a *snail* mutant and, therefore, contribute less to the overall distribution of local tension configurations. Indeed, no increase in the fraction of tension bridges is observed in this mutant (orange line in *Figure 8D*).

## Discussion

### Cortical tensions drive and constrain tissue deformations on the cellular scale

We have developed a novel perspective on tissue dynamics based on the principle of force balance. Central to our approach is the balanced network of active forces generated and transmitted by cells. In epithelia where contractility in the adherens-junctional cortex is the strongest source of stress, the force network takes the form of a triangulation in tension space (*Noll et al., 2017*) which fixes the angles at tri-cellular vertices in the tissue. Tissue deformations are driven by the adiabatic transformation of this force-balance geometry, allowing the epithelium to rearrange like a fluid while supporting internal tension like a solid. Alternatively, one can think of this behavior as a form of plasticity where the internal 'reference' structure of the material can be actively remodeled, resulting in spatial displacement of material points. However, the focus on the force balance geometry associated with the internal structure is more useful as it holds the key to understanding possible mechanisms of active control. We find two mechanically distinct modes of tissue deformation: a cortical-tension-driven mode and an isogonal (angle preserving) mode that is unconstrained by the cortical force-balance geometry. The former represents active remodeling of interfaces, e.g., by myosin recruitment, while the latter accounts for passive tissue deformations that are controlled by external (non-local) forces and cell shape elasticity. The *tension–isogonal decomposition* quantifies the contributions of these two modes in experimental data based on cell geometry alone. This allows us to disentangle whether deformations are due to locally or non-locally generated forces, i.e., whether they are active or passive. Importantly, tension–isogonal decomposition is based on the same assumptions as tension inference which has been validated extensively in previous literature (*Ishihara et al., 2013*; *Kong et al., 2019*; *Noll et al., 2020*; *Cheikh et al., 2022*).

In the early *Drosophila* embryo, we observe both active and passive cell rearrangements (T1 events). Generally, both modes – tension-driven and isogonal – interact and contribute to tissue deformations at the same time. We have found that ventral furrow invagination isogonally stretches the lateral ectoderm which increases the tension threshold for active T1s there. This predicts that the local relative tension threshold will be lower (namely $\sqrt{2}$) in mutants that lack a ventral furrow (*twist* and *snail Gheisari et al., 2020*). Indeed, analysis of a *snail* mutant embryo confirmed this prediction and thus validates that tension–isogonal decomposition captures the interplay of internal and external forces acting on a tissue.

This predicts that the local relative tension threshold will be lower (namely $\sqrt{2}$) in mutants that lack a ventral furrow (*twist* and *snail Gheisari et al., 2020*). Indeed, analysis of a *snail* mutant embryo confirmed this prediction and thus validates that tension–isogonal decomposition captures the interplay of internal and external forces acting on a tissue.

### Internal and external contributions to germ band extension

Where the forces driving tissue deformations during morphogenesis originate is a fundamental question of developmental biology. In the context of *Drosophila* gastrulation, it has been has been intensively debated whether the germ band elongates due to internally generated stresses (*Irvine and Wieschaus, 1994*; *Bertet et al., 2004*; *Rauzi et al., 2008*; *Butler et al., 2009*; *Gustafson et al., 2022*; *Streichan et al., 2018*) or due to external pulling by the posterior midgut invagination (*Collinet et al., 2015*; *Lye et al., 2015*; *Bailles et al., 2019*; *Gehrels et al., 2023*). Experimental evidence shows that both processes contribute, making a nuanced, quantitative analysis necessary. We, therefore, re-analyzed microscopy data from previously published cauterization and mutant experiments

(*Collinet et al., 2015*) to quantify tissue flow (see Appendix 6). The results from this quantitative analysis indicate that forces generated in the germ band contribute significantly to tissue flow. This conclusion is further supported by the observations from mutants where posterior midgut invagination is disrupted (*fog*, *torso-like*, *scab*, *corkscrew*, *ksr*). In these mutants, the germ band buckles forming ectopic folds (*Zusman and Wieschaus, 1985*; *Parks and Wieschaus, 1991*) or twists into a corkscrew shape (*Perkins et al., 1992*; *Smits et al., 2023*) as it extends, pointing towards a buckling instability characteristic of internally driven extensile flows (*Liu et al., 2006*; *Senoussi et al., 2019*). This suggests that the main effect of PMG invagination on the germ band lies not in creating pulling forces, but rather in 'making room' to allow for its orderly extension.

However, these tissue-scale observations only provide circumstantial evidence for internally driven GBE. To conclusively settle the debate, evidence from the cell scale – where the forces are generated – is needed. Such evidence is provided by our tension–isogonal analysis, which yields a fine-grained picture with various regions of active and passive deformation (see *Figure 6B and B'*). It clearly shows that tissue deformation in the germ band is driven by internal remodeling of tensions and, therefore, active. In stark contrast, the amnioserosa, and the tissue just anterior and lateral of the invaginating posterior midgut deforms passively.

## Cells orchestrate tissue flow by self-organizing in tension space

The internally driven nature of germ-band elongation flow immediately raises the question of how force generation is coordinated across cells to drive coherent tissue flow. In other words, how can the local behavior of cells orchestrate global tissue flow? Cells exert active stresses on each other and, at the same time, constantly sense their mechanical environment (*Heisenberg and Bellaïche, 2013*; *Collinet and Lecuit, 2021*; *Pinheiro and Bellaïche, 2018*). Mechanical stresses and strains can propagate over long distances and contain information about the tissue geometry (e.g. in the form of hoop stresses *Lefebvre et al., 2023*). In an epithelium dominated by cortical tensions, this mechanical environment forms a 'tension space,' linked to physical space via force balance. Tension space takes the form of a triangulation which allows an intuitive visualization. The angles in the cell array are fixed by complementary angles in the tension triangulation. Thus, the tensions are geometric dials cells can directly sense and control.

We found that cells can control their configuration in tension space by defining the local dynamics of cortical tension. In experimental data from gastrulating *Drosophila* embryos, we identified two distinct behaviors: (*i*) amplification of tension on the interfaces that are already most tense, suggestive of positive tension feedback (observed in lateral ectoderm) and (*ii*) apparent tension homeostasis (observed in the amnioserosa). Mechanical homeostasis is found in various systems and organisms (*Humphrey, 2008*; *Latorre et al., 2018*; *Stamenović and Smith, 2020*). We hypothesize that tension homeostasis allows the amnioserosa to undergo large cell deformations while maintaining tissue integrity (*Latorre et al., 2018*; *Jodoin et al., 2015*). Tension feedback, on the other hand, continuously modifies local force balance, driving the change in cell and tissue geometry. To the extent that cortical responses can be controlled by spatially modulated gene expression, evolution thus has the means to define a program of non-trivial spatiotemporal dynamics of tissue during morphogenesis. Evidence for positive feedback is provided by the nonlinearly increasing inferred tension on contracting interfaces (*Figure 2D*) and by laser ablation experiments in earlier work (*Blankenship et al., 2006*). The underlying molecular mechanisms are yet to be identified and might rely on the catch-bond behavior of myosin *Veigel et al., 2003*; *Laakso et al., 2008* or mechano-sensitive binding of $\alpha$-catenin (*Barry et al., 2014*; *Mei et al., 2020*).

We have found that T1 events can be explained quantitatively through a simple geometric criterion that defines a 'T1 threshold' in terms of the local tension configuration and the local isogonal strain. A minimal mathematical model of an intercalating cell quartet demonstrates how positive tension feedback can drive the tensions towards this T1 threshold and thus generate active T1s. In contrast, passive T1s result from external forces changing the isogonal modes until a cell neighbor exchange occurs, while tensions (and thus interface angles) remain constant.

Our mathematical model for an intercalating cell quartet also sheds light on the long-standing puzzle of how interfaces extend after a neighbor exchange, even though cortical myosin can only exert contractile forces. We show that no additional active mechanism is necessary. Our minimal model captures interface extension as a purely passive process. In our model, interface extension

during both active and passive intercalations is driven by the tension on adjacent interfaces which exceeds the low active contractility of the new edge. The low myosin level on the emerging interface is predicted by a simple model for the myosin 'handover' during a neighbor exchange. In tissue-scale simulations (*Claussen et al., 2024*), the model reproduces experimental observations (interface extension even if tissue extension is blocked, generation of irregular cell shapes) that were previously taken as evidence for actively driven interface extension (*Collinet et al., 2015*; *Vanderleest et al., 2022*).

Our findings on the dynamics of interface length and tension during active T1s (*Figure 2D*) are similar to another recently proposed model (*Sknepnek et al., 2023*). This model extends the classical 'vertex model' with area–perimeter elasticity by active cortical contractility. Like in our model, active T1s are driven by a positive feedback loop of cortical tension. Since our model is formulated in adiabatic force balance, it requires fewer parameters than the one of *Sknepnek et al., 2023* and might be obtained from the latter in the limit of fast mechanical relaxation and dominant active tensions.

In other recent theoretical work, *Drosophila* extension germ band has been interpreted as resulting from tissue *fluidization* (*Wang et al., 2020*), i.e., the loss of resistance to shear stress due to the vanishing of cortical tensions (*Farhadifar et al., 2007*; *Yan and Bi, 2019*). However, laser ablation experiments (*Ma et al., 2009*; *Collinet et al., 2015*) show that cortical tensions remain finite during the morphogenetic process. In fact, interfaces under vanishing would generically buckle into a wiggly shape (as observed, for instance, in the amnioserosa during dorsal closure *Tah et al., 2023* and in mutants with increased medial myosin *Latorre et al., 2018*). This is not observed in the *Drosophila* ectoderm during gastrulation where interfaces instead appear taught straight, serving as a simple visual indicator of finite tension. We have, therefore, investigated tissue mechanics in the regime where the tissue internally generated tension at all times and hence is a solid rather than a fluid. Tissue flow occurs as a result of internally driven *plastic rearrangements*, and does not require or imply tissue fluidization. The capacity of epithelial tissue to internally rearrange in a controlled manner – which may be thought of as *active plasticity* – falls beyond the fluidization paradigm. The results of our analyses on the cell and tissue scale corroborate this point of view. We return to the question of fluid vs solid behavior in the companion paper (*Claussen et al., 2024*).

## Order in local tension configurations coordinates active T1s

We found a striking alternating pattern of cortical tensions that explains how T1s are coordinated between neighboring cells. The prevalence of this organizing motif is quantified by an order parameter

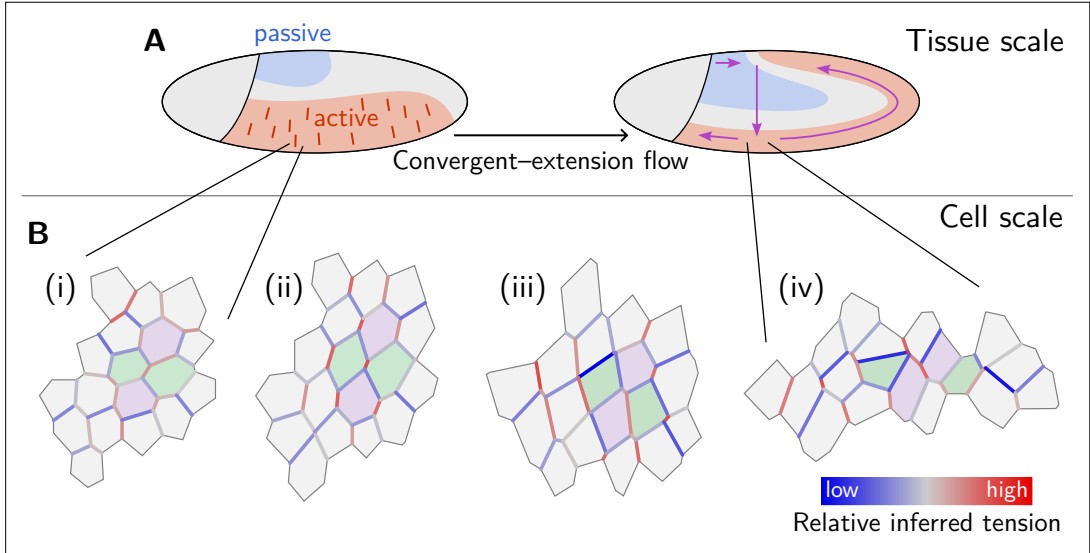

**Figure 9.** Mechanical coordination of convergent extension flow on the tissue and cell scale. (**A**) Dorso-ventral patterning of mechanical properties and tension anisotropy (red lines) organize and orient tissue flow. (**B**) A cell scale pattern of tensions coordinates active cell intercalations that drive convergent extension of the lateral ectoderm. From relatively uniform but weakly anisotropic initial tensions (i), an alternating pattern of high and low tensions emerges (ii). Subsequently, high-tension interfaces fully contract thereby driving parallel active T1 transitions (iii). As T1s proceed, the order in the tension configurations is lost, and convergent extension flow cedes (iv).

for LTC. LTC order precedes intercalations in lateral ectoderm and the loss of LTC order correlates with the slowdown of tissue flow at the end of GBE. This raises two important questions: First, how does LTC order emerge? Second, is the loss of LTC order the cause for the termination GBE? These questions are addressed in a companion paper (*Claussen et al., 2024*) using a tissue-scale mathematical model based on the quartet-scale model introduced here. We show that LTC order is driven by positive tension feedback and requires an initial regular hexagonal packing of cells. Intercalations disrupt this regular hexagonal packing and thus leads to the loss of LTC order as GBE progresses (see illustration in *Figure 9B*). Together with the degradation of coherent tension orientation, loss of LTC order causes T1s to become incoherent and, therefore, ineffective at driving tissue flow. Taken together, this implies that convergent extension flow is self-limiting and that the final tissue shape is encoded in the initial degree hexagonal order and tension anisotropy. Moreover, we predict that disrupting the hexagonal packing of nuclei prior to cellularization will prevent the emergence of LTC order and thus cause slower GBE.

In addition to facilitating rapid and efficient convergent extension flow, the coordination of T1s also helps preserve the genetic patterns that are set up before germ band extension (*Nüsslein-Volhard and Wieschaus, 1980*) (see Appendix 1.4 and *Figure 1—figure supplement 3*). These patterns have been found to have sufficient fidelity to resolve the positions of individual cells along the AP axis (*Dubuis et al., 2013*) and are, therefore, highly susceptible to disorder resulting from incoherent cell rearrangements. This suggests that high-fidelity patterning and highly coordinated cell rearrangements might have co-evolved to maximize the speed of development.

## Genetic patterning and initial anisotropy

Because the *Drosophila* embryo is a closed surface, convergent extension of the germ band must be compensated by an orthogonal deformation of the dorsal tissue, the amnioserosa (see *Figure 9A*). To achieve that, the mechanical properties of the tissue must be modulated along the dorsoventral axis: Positive tension feedback drives active rearrangements in the lateral ectoderm, and tension homeostasis in the amnioserosa allowing it to yield to external forces while maintaining tissue integrity. This highlights the importance of the DV-axis patterning. Recent work (*Gustafson et al., 2022*) has shown that mechanical feedback loops in the embryo are patterned along the DV axis. Notably, the DV patterning system is conserved across species (*Holley and Ferguson, 1997*). Our work also highlights a second interaction between tissue flow and DV patterning: because the coherent T1s we observe do not mix cells in the tissue (*Figure 1A*), distinct DV fates (e.g. neural and surface ectoderm) remain clearly demarcated, a clear biological necessity.

In addition to the DV gradient, tension anisotropy along the DV axis is required to orient the GBE flow along the AP axis. This anisotropy might be due to mechanics (pulling of the ventral furrow [*Gustafson et al., 2022*] and hoop stresses resulting from ellipsoidal embryo geometry [*Lefebvre et al., 2023*] since we observe anisotropy already prior to VF formation), or indeed due to AP-striped genetic patterning (*Butler and Wallingford, 2017*; *Lavalou et al., 2021*). Thanks to the positive tension feedback, a weak initial anisotropy is sufficient to bias the direction of tissue elongation. This explains why *twist* and *snail* mutants, which have significantly reduced initial myosin anisotropy, still extend their germ band, albeit at a slower rate (*Gustafson et al., 2022*).

Our findings show how genetic patterning and self-organization are interconnected during GBE. Genetic patterning provides tissue scale input by demarcating distinct tissue regions with different mechanical properties. Self-organization via positive tension feedback orchestrates myosin contractility on the cellular scale and drives T1s. In the companion paper (*Claussen et al., 2024*), we show that mechanical self-organization via tension feedback also allows cells to coordinate their behaviors (such as active T1s) on the tissue scale to give rise to coherent morphogenetic flow. As of yet, no genetic mechanism has been found that produces cell scale coordination – as manifested, for instance, in the alternating pattern of high and low tensions (*Figure 8A*, center) – from predetermined genetic patterns alone. Moreover, many interfaces in the lateral ectoderm that undergo T1 transitions at late stages of GBE are initially oriented along the AP, rather than the DV axis and only rotate into DV alignment briefly before they collapse (see *Figure 1—figure supplement 2*). This shows that an initially set-up genetic pattern is not sufficient to explain the coordination of cell rearrangements, in line with evidence from tissue scale analysis (*Lefebvre et al., 2023*). Instead, the emerging overarching picture is that of morphogenesis driven by an interplay of bottom-up local self-organization controlled by

top-down genetic patterning that sets initial conditions and modulates parameters on the tissue scale (*Schweisguth and Corson, 2019*; *Lenne et al., 2021*). Indeed, this interplay has recently been shown to underlie posterior midgut invagination, where genetic patterning initiates and channels a wave that mechanically propagates due to a local feedback mechanism acting on myosin (*Bailles et al., 2019*).

Going forward, we expect that geometric insight into tension space will provide a new perspective on many different systems (*Shindo, 2018*), for example, *Xenopus* neural tube formation (*Butler and Wallingford, 2018*), amniote primitive streak formation (*Voiculescu et al., 2007*; *Saadaoui et al., 2020*; *Rozbicki et al., 2015*), wing disk elongation (*Dye et al., 2021*; *Dye et al., 2017*), the sea urchin archenteron (*Hardin and Weliky, 2019*), or kidney tubule elongation (*Lienkamp et al., 2012*). Furthermore, it will be interesting to study intercalations driven by actin 'protrusions' (*Weng et al., 2022*): they too are governed by cortical force balance on the cellular level, making tension inference and the tension–isogonal decomposition applicable.

## Acknowledgements

We thank Noah Mitchell, Sebastian Streichan, and Arthur Hernandez for stimulating discussions and careful reading of the manuscript. We would also like to thank the anonymous reviewers for their excellent feedback on the manuscript. FB acknowledges the support of the Gordon and Betty More Foundation post-doctoral fellowship (under grant #2919). NHC was supported by NIGMS R35-GM138203 and NSF PHY:2210612. MFL was supported by NSF Career grant No. PHY-2047140. EFW acknowledges support from Howard Hughes Medical Institute. BIS acknowledges support via NSF PHY:2210612.

## Additional information

### Funding

| Funder | Grant reference number | Author |
| --- | --- | --- |
| National Science Foundation | PHY:2210612 | Boris I Shraiman |
| National Institute of General Medical Sciences | R35-GM138208 | Nikolas H Claussen |
| Gordon and Betty Moore Foundation | 2919 | Fridtjof Brauns |
| National Science Foundation | PHY-2047140 | Matthew F Lefebvre |

The funders had no role in study design, data collection and interpretation, or the decision to submit the work for publication.

### Author contributions

Fridtjof Brauns, Nikolas H Claussen, Conceptualization, Software, Formal analysis, Investigation, Methodology, Writing – original draft, Writing – review and editing; Matthew F Lefebvre, Investigation, Writing – review and editing; Eric F Wieschaus, Conceptualization, Methodology, Writing – review and editing; Boris I Shraiman, Conceptualization, Formal analysis, Funding acquisition, Methodology, Writing – original draft, Writing – review and editing

### Author ORCIDs

Fridtjof Brauns ⓘ https://orcid.org/0000-0002-6108-9278
Nikolas H Claussen ⓘ https://orcid.org/0000-0002-9020-6437
Matthew F Lefebvre ⓘ https://orcid.org/0000-0002-9590-4293
Eric F Wieschaus ⓘ https://orcid.org/0000-0002-0727-3349
Boris I Shraiman ⓘ https://orcid.org/0000-0003-0886-8990

Reviewer #2 (Public review): https://doi.org/10.7554/eLife.95521.3.sa1
Reviewer #3 (Public review): https://doi.org/10.7554/eLife.95521.3.sa2

Author response https://doi.org/10.7554/eLife.95521.3.sa3

## Additional files

### Supplementary files
• MDAR checklist

### Data availability
Mathematica code used to analyze experimental data is available in the repository https://github.com/f-brauns/GastrulationTensionIsogonal/tree/0f8f55f (copy archived at *Brauns, 2024*). Simulation code (Python) is available in the repository https://github.com/nikolas-claussen/CE_simulation_public (copy archived at *Claussen, 2024a*). Whole embryo cell segmentation and tracking data for a WT embryo (strain: Tub67a < CAAX-mCherry < sqh3'UTR.attp2. / TM3,Sb) was obtained from the repository https://doi.org/10.6084/m9.figshare.18551420.v3, deposited with *Stern et al., 2022*. We analyzed the dataset with id number 1620 since it had the highest time resolution (15 s) and covered the longest time period (50 min), starting ca. 7 min before the onset of ventral furrow invagination. For the analysis of the snail mutant, we used data from the *Drosophila* morphodynamic atlas (*Mitchell et al., 2022*). This data is available on the data repository https://doi.org/10.25349/D9WW43. We used recording No. 202207111700 (strain: halo snail[IIG05]; CAAX-mCherry). For cell segmentation and tracking, we used the Fiji plugin Tissue Analyzer (*Aigouy et al., 2016*) and custom Mathematica code. For analysis of T1 transitions and isogonal strain, we excluded regions that form folds. Note that in recording No. 202207111700, one dorsal fold ectopically extends to the ventral side.

The following previously published datasets were used:

| Author(s) | Year | Dataset title | Dataset URL | Database and Identifier |
| --- | --- | --- | --- | --- |
| Stern T, Shvartsman SY, Wieschaus EF | 2022 | Deconstructing Gastrulation at Single-Cell Resolution | https://doi.org/10.6084/m9.figshare.18551420.v3 | figshare, 10.6084/m9.figshare.18551420.v3 |
| Mitchell N, Lefebvre M, Jain-Sharma V, Claussen N, Raich M, Gustafson H, Bausch A, Streichan S | 2022 | Morphodynamic atlas for *Drosophila* development | https://doi.org/10.25349/D9WW43 | Dryad Digital Resipotary, 10.25349/D9WW43 |

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

# Appendix 1

## Tissue tectonics

### 1.1 Tissue tectonics analysis

Based on segmented and tracked cells, the tissue strain rate and cell deformation rate can be calculated via the methods described in *Blanchard et al., 2009*; see also *Guirao et al., 2015* and *Merkel et al., 2017*. In brief, the local tissue strain rate is calculated from the centroid positions $\mathbf{c}_i$ of the cells in a small neighborhood $N$ of a cell (we use nearest neighbors). One looks for a linear transformation that maps the positions of the cell centroids at one-time point to those at the next one. Because the resulting equations are overdetermined, an approximate solution minimizing the error is obtained using the least-squares algorithm. The linear transformation is then further decomposed into a symmetric part, describing local shear and area changes, and an antisymmetric part, describing the tissue rotation. Because the *Drosophila* ectoderm is a surface in three-dimensional space, we need to calculate these quantities in the local tangent plane. The cell deformation rate is calculated from the rate of change of the cell's moment of inertia tensor in the frame that co-rotates with the tissue.

*Figure 1—figure supplement 1A–B'* show the tissue and cell deformation rates during VF invagination (*Figure 1—figure supplement 1A, A'*) and GBE (*Figure 1—figure supplement 1B, B'*). *Figure 1—figure supplement 1C* shows the cell elongation as measured by the Beltrami coefficient $\mu = |a - b|/(a + b)$, where $a$ and $b$ are the half-axes of a best-fit ellipse to the cell.

### 1.2 Local tangent plane and coordinate frames

*Figure 5Figure 6*

The analysis of local tissue geometry (including tissue tectonics, tension inference, and tension–isogonal decomposition) is performed in the local tangent plane. The local tangent plane is determined by the local normal vector $\mathbf{n}$ which we find by a singular value decomposition of the point set of interest (vertex coordinates for a single cell, cell centroids for a patch of cells). The normal vector is the basis vector corresponding to the smallest singular value.

Depending on the context, we use three different coordinate bases in the local tangent plane: (**i**) The 'Eulerian' (or 'lab frame') basis for a cell $i$ is formed by the azimuthal vector $\mathbf{e}_{i,\mathrm{DV}}^{\mathrm{E}}$ along the DV axis and its orthogonal complement $\mathbf{e}_{i,\mathrm{AP}}^{\mathrm{E}} = \mathbf{n} \times \mathbf{e}_{i,\mathrm{DV}}^{\mathrm{E}}$, along the AP axis. The Eulerian basis vectors are illustrated in *Appendix 1—figure 1* on the left (0 min).

(*ii*) A 'Lagrangian' (or 'tissue frame') basis which co-rotates with the tissue is used to quantify the DV component of the isogonal deformation tensor (*Figure 5C*) and the orientation of collapsing interfaces (*Figure 1—figure supplement 2*). This co-rotating frame is defined by tracking a patch of cells neighboring cell $i$ and fitting a linear transformation $A_i(t)$ to the displacement of cell centroids relative to the patch's center of mass. The linear transformation is constrained to transform the initial into the final normal vector $A_i.\mathbf{n}_i(0) = \mathbf{n}_i(t)$. A polar decomposition $A_i(t) = S_i(t).R_i(t)$ then yields the local rotation matrix $R_i(t)$ for the tissue patch. The co-rotated basis vectors for cell $i$ are obtained by applying $R_i$ to the cell's Eulerian basis vectors at time zero: $\mathbf{e}_{i,\mathrm{AP}}^{\mathrm{L}}(t) = R(t).\mathbf{e}_{i,\mathrm{AP}}^{\mathrm{E}}(0)$, $\mathbf{e}_{i,\mathrm{DV}}^{\mathrm{L}}(t) = R(t).\mathbf{e}_{i,\mathrm{DV}}^{\mathrm{E}}(0)$. The Lagrangian basis vectors are illustrated in *Appendix 1—figure 1* on the right (35 min). (*iii*) For the tension–isogonal decomposition of intercalating cell quartets (*Figure 4C and C'* and *Figure 4—figure supplement 1*), we define a co-moving orthogonal basis based on the cell centroids: The basis vectors are defined as the pair of orthogonal vectors that best fit the vectors that connect opposite cells in the quartet. Before the neighbor exchange, the first basis vector is aligned with the two cells that touch and the second basis vector is aligned with the cells that don't touch. After the neighbor exchange, the roles are flipped. This ensures that the basis vectors vary smoothly through the neighbor exchange.

### 1.3 Identification and classification of intercalation events

Cell interface collapse and emergence events can be identified by tracking cell contacts. We start by identifying all pairs of cells that share an interface at some time point. For each pair, one can then identify the time periods where the cells are in contact. This allows one to identify interface collapse and emergence events. We exclude cases where a cell pair loses contact because one of the cells invaginates or divides and cases where cells come into contact because cells in between them invaginate.

Some cell pairs transiently lose or gain contact. For the interface collapse rate shown in *Figure 1E*, we only account for the first collapse event for cell pairs that are not in contact at the last frame, i.e., for those that *permanently* lose contact. Similarly, for interface emergence events, we only account for the last event where a pair of cells not in contact at the first frame comes into contact.

## Spatial distribution of interface collapses

From the tracked interface collapses, we can count the number of interface collapses that each cell is involved in, giving us a spatially resolved map of intercalations (*Figure 1—figure supplement 1D*). In the lateral ectoderm (regions 4–8 in *Figure 2—figure supplement 1A*), we find an average of 1.29 interface collapses per cell. From this, we can estimate the total tissue extension by cell rearrangements. In a hexagonal lattice, synchronous T1s cause one interface collapses per cell and elongate the tissue by a factor $\sqrt{3}$, assuming that the cells do not elongate (The tissue also contracts by a factor $1/\sqrt{3}$ along the orthogonal axis). For 1.29 interface collapses per cell, therefore, cause tissue elongation by a factor $1.29 \times \sqrt{3} \approx 2.23$, in good agreement with the total elongation of the germ band (*Figure 1D*).

In contrast to the lateral ectoderm where cells don't elongate coherently, cell elongation contributes significantly to tissue deformation near the dorsal pole (see *Figure 1—figure supplement 1C*). This is a generic feature of passive T1s driven by isogonal deformations (see *Figure 4—figure supplement 1D*).

## Orientation of collapsing interfaces

The convergent extension flows in the lateral ectoderm and near the dorsal pole (amnioserosa) are oriented perpendicular to each other (see *Figure 1—figure supplement 1*). We, therefore, also expect perpendicular orientation of the collapsing interfaces in these two regions. We quantify the orientation of collapsing edges at the blastoderm stage (i.e. before the onset of tissue flow, $t = 0\,\mathrm{min}$) as shown in *Figure 1—figure supplement 2A and C*. In the ectoderm, interfaces that collapse before $t = 42\,\mathrm{min}$, i.e., within the first ca. 15 min of GBE are predominantly DV oriented at $t = 0\,\mathrm{min}$ (see *Figure 1—figure supplement 2B*). By contrast, interfaces near the dorsal pole are predominantly oriented along AP. The orientation of interfaces in the lateral ectoderm that collapse after $t = 42\,\mathrm{min}$ is initially not strongly biased along the DV axis *Figure 1—figure supplement 2C and D*. However, during GBE, these interfaces progressively align with the DV co-rotated DV axis before they eventually start contracting (see *Figure 1—figure supplement 2C'* and D; the coordinate frame co-rotating with the tissue is described in Appendix 1.1).

## Tracking quartets and identifying rosettes

The above procedure for identifying intercalation events is very fast and easy to implement. However, it does not distinguish between T1s and higher-order intercalation events (rosettes). Moreover, it cannot easily be determined which interface collapse corresponds to which interface emergence.

We, therefore, implemented an alternative algorithm to identify intercalations, by tracking quartets of cells. This is possible by identifying cycles of length four in the neighborhood graph of the cell array at each time point. We exclude cases where the four cells surround one or more cells that are not part of the cycle and cases where one of the cells is surrounded by the three others. Notably, the quartets don't necessarily exist for the entire time series. Namely, when one of the 'inner' interfaces in the quartet, other than the central interface, collapses, the quartet is destroyed.

A T1 event happens when the opposite pairs of cells in the quartet exchange neighbors (i.e. one pair loses contact and the other one gains it). In some cases, the neighbor exchange is not instantaneous, but instead, there is a fourfold vertex in the center of the quartet for an extended period. Cases where the central interface and one of its neighbors are simultaneously shorter than a threshold length ($0.5\,\mu\mathrm{m}$) are classified as rosettes and not counted as T1s. The threshold value $0.5\,\mathrm{m}$ corresponds to the average vertex-positions fluctuations on the timescale of 1 min. Higher order rosettes can be 'decomposed' into overlapping fivefold rosettes and, therefore, can be identified by finding such overlaps.

Using this procedure, we identified 2231 T1 events, 498 fivefold rosettes, 40 sixfold rosettes, 4 sevenfold rosettes, and no higher order rosettes (see *Figure 1E'*). Because the number of rosettes is small compared to the number of T1s, we have focused on T1 events for the subsequent analysis. Expressed in terms of a participation ratio of interface contractions in T1s vs rosettes, we find a ratio

$2231 : (2 \cdot 498 + 3 \cdot 40 + 4 \cdot 4) \approx 2 : 1$, matching observations from previous studies (*Farrell et al., 2017*; *Stern et al., 2022*).

## 1.4 Cell rearrangements preserve AP stripe patterns

During the early development of the *Drosophila* embryo a hierarchy of genetic patterns is set up before morphogenetic flow starts. This hierarchy culminates in the striped expression of the pair-rule genes which lay out the segmented body plan of the larva (*Nüsslein-Volhard and Wieschaus, 1980*). Detailed quantification of the gene expression profiles has revealed a remarkable fidelity of this patterning system which is sufficient to identify each cell's position along the AP axis (*Dubuis et al., 2013*). Germ band extension starts after this pattern has been established, raising the question of how cell rearrangement affects the pattern fidelity. Cell tracking allows us to test this question computationally. To visualize the effect of cell rearrangements on an AP pattern, we assign colors to cells according to their initial AP position, coarse-grained into ca 2 cell-wide stripes (*Figure 1—figure supplement 3A*, 0 min). Rendering the cells with these colors at a later time shows sharp boundaries between stripes with few outliers, revealing that the stripe pattern is locally preserved with high fidelity (*Figure 1—figure supplement 3A*, 40 min). On large scales, the pattern is distorted because the posterior germband curves over the posterior pole as it elongates (*Figure 1A*).

The local preservation of pattern fidelity is in line with the highly choreographed active T1s in the germ band (*Figure 1C*). As a reference, we simulated purely passive cell rearrangement driven by externally imposed strain (*Figure 1—figure supplement 3B*). In brief, cells were simulated as foam bubbles with constant interfacial tension and area-elasticity. The externally imposed strain was implemented by forces applied to the top and bottom boundaries. Notably, the stripe boundaries after twofold elongation appear significantly more disrupted than in the embryo, suggesting that coordination of T1s in the germ band helps preserve pattern fidelity. Simulations were carried out as described in *Claussen et al., 2024*.

To quantify the local pattern fidelity we define the relative AP position for cell pairs ($ij$) as

$$d_{ij}^{\mathrm{AP}}(t) := \mathbf{e}_{i,\mathrm{AP}}^{\mathrm{L}}(t) \cdot (\mathbf{c}_i(t) - \mathbf{c}_j(t)), \tag{4}$$

where $\mathbf{c}_i(t)$ is the centroid position of cell $i$ at time $t$ we project onto the advected (Lagrangian) local basis vectors $\mathbf{e}_{i,\mathrm{AP}}^{\mathrm{L}}(t)$ defined inAppendix 1.2 to measure relative AP positions in relation to the original blastoderm stage coordinate frame. From the $d_{ij}^{\mathrm{AP}}(t)$ we define the correlation coefficient

$$C(t) = \langle d_{ij}^{\mathrm{AP}}(0) d_{ij}^{\mathrm{AP}}(t) \rangle, \tag{5}$$

where $\langle \ldots \rangle$ denotes the average over all pairs that are neighbors at time $t$. The correlation $C(t)$ plotted against the number of interfaces lost per cell measures how well the local gradient of an AP pattern is preserved during convergent extension (*Figure 1—figure supplement 3C*). For the anterior germ band, we find significantly less degradation of fidelity compared to the passive stretching simulation. The posterior germ band, which is distorted as it curves over the posterior pole, exhibits a similar degradation rate as the passive reference case. Notably, the rate of pattern degradation (i.e. the slope of the curve) increases towards the end in the anterior germ band, reaching the same degradation rate as the posterior germ band and passive reference. This is consistent with the loss of coherence of T1s towards the end of germ band extension which likely results from the loss of large-scale tension anisotropy (*Figure 5C and D*) and LTC order (*Figure 8C and D*).

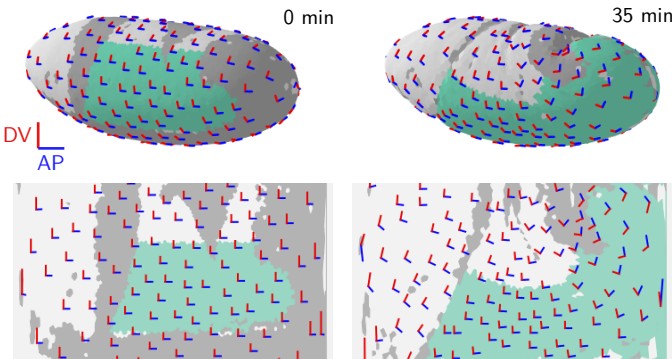

**Appendix 1—figure 1.** Local basis vectors aligned with the initial AP (blue) and DV (red) directions. On the right (35 min), the basis vectors have been advected with the tissue flow, i.e., they are the local Lagrangian basis vectors. Note that in the pullback maps, the basis vectors appear distorted near the poles because the projection is not angle- or length-preserving. In figures showing the magnitudes and orientations of anisotropic quantities such as strain and tension this distortion has been removed (*Figure 5*, *Figure 6*, *Figure 1—figure supplement 1*, *Figure 6—figure supplement 1* and *Figure 7—figure supplement 1*).

## Appendix 2

### Local tension inference

Consider the triplet of cells that meet at a tri-cellular vertex and label them $i = 1, 2, 3$. Denote the unit vectors pointing along the three interfaces that meet at the vertex by $\mathbf{e}_{ij}$, where $i, j$ are the labels of the pair of cells that share the respective interface. Force balance implies that

$$T_{12}\mathbf{e}_{12} + T_{23}\mathbf{e}_{23} + T_{31}\mathbf{e}_{31} = 0, \tag{6}$$

where $T_{ij}$ are the cortical tensions. Multiplying the above equation with $\hat{\mathbf{e}}_{12}$ and $\hat{\mathbf{e}}_{23}$ (where $\hat{\mathbf{e}}$ denotes the vector orthogonal to $\mathbf{e}$ and with the same magnitude) yields two equations

$$T_{23} \sin \alpha_{123} - T_{31} \sin \alpha_{312} = 0,$$
$$T_{12} \sin \alpha_{123} - T_{31} \sin \alpha_{231} = 0, \tag{7}$$

Here, $\alpha_{ijk}$ denotes the angle subtended by the interfaces $(ij)$ and $(jk)$. An alternative, more geometric way to arrive at these equations is by applying the law of sines to the tension triangle, using that its angles are complementary to those at the vertex and $\sin(\alpha - \pi) = \sin(\alpha)$.

*Eqs. (7)* equations do not constrain the overall level of tension. We fix this by setting the average tension to 1 which gives the solution These equations are solved by

$$T_{12} = \mathcal{N} \sin \alpha_{231}, \quad T_{23} = \mathcal{N} \sin \alpha_{312}, \quad T_{31} = \mathcal{N} \sin \alpha_{123}, \tag{8}$$

with the normalization factor $\mathcal{N} = \sum_{(ijk)} \sin \alpha_{ijk}$.

The above calculation can be generalized to an arbitrary number of vertices. Notably, the equations become overconstrained once the set of vertices contains entire cells (*Noll et al., 2017*). This is because force balance implies that the tension vectors $\mathbf{T}_{ij} = T_{ij}\mathbf{e}_{ij}$ form a triangulation (see below).

Here, we limit ourselves to local tension inference for the central interface in a cell quartet (see *Figure 2A*). In this case, the equations are not over-constrained. Labeling the cells 1–4, the equations read

$$T_{23} \sin \alpha_{123} - T_{13} \sin \alpha_{312} = 0,$$
$$T_{12} \sin \alpha_{123} - T_{13} \sin \alpha_{231} = 0,$$
$$T_{34} \sin \alpha_{143} - T_{13} \sin \alpha_{314} = 0,$$
$$T_{14} \sin \alpha_{143} - T_{13} \sin \alpha_{431} = 0, \tag{9}$$

We solve these equations under the normalization condition that the average of the 'outer tensions' $T_{12}, T_{23}, T_{34}$ and $T_{14}$ is 1. The inferred value for 'inner' cortical tension $T_{13}$ then is that relative to the average tension of its neighbors. We call this the *relative tension* of an interface throughout the manuscript.

The use of local tension inference is justified because we are interested in local cell behaviors, namely rearrangements. Tension inference is also most robust on the local level since this is where force balance, the underlying physical determinant of the link between mechanics and geometry, resides. In global tension inference, spurious large-scale gradients can appear when small deviations from local force balance accumulate over large distances. For this reason, we do not try to infer tension gradients in the magnitude of tension across the embryo, e.g., between the amnioserosa and the lateral ectoderm.

### Analysis of collapsing and emerging interfaces

For the analysis presented in *Figure 2D* and *Figure 2E*, collapsing and emerging interfaces were identified separately as described in Appendix 1.3. All events before $12\,\mathrm{min}$ were excluded from the analysis.

When an interface is short, the interface angles become highly susceptible to fluctuations in the vertex positions. We, therefore, excluded all time points where the central interface or one of its neighbors is shorter than a threshold length ($0.75\,\mu\mathrm{m}$). This value was chosen because the vertex positions fluctuate about $0.5\,\mu\mathrm{m}$ between time points. These fluctuations are partially due

to intrinsic fluctuations in cortical tension and partially result from limitations in vertex localization during segmentation because of limited optical resolution.

The time series of interface length and relative tension are then aligned such that the interface collapse (respectively emergence) is at $t = 0$. *Figure 2—figure supplement 1* shows the pooled data resolved by the position of the quartet along the DV axis.

## Appendix 3

### Tension–isogonal decomposition

Because the angles in the tension triangulation are complementary to the angle at vertices in the cell array, one can find a unique tension triangulation (up to a global scale factor) for a given physical cell array in force balance (see dashed gray arrow in *Figure 4—figure supplement 1A*). However, the reverse is not true. For a given tension triangulation, there is a continuous family of cell arrays that share the same tension triangulation and that are related by *isogonal* deformations which change the interface lengths collectively while preserving the angles. (Mathematically, there are fewer triangulation vertices than cell vertices, so the triangulation leaves one isogonal mode degree of freedom per cell undetermined. At the end of this section, will define an 'isogonal potential' that parametrizes the isogonal modes. Specifically, we find that a quadratic potential corresponds to a uniform pure shear of the cell centroids that preserves the angles at vertices).

The isogonal degrees of freedom reflect tissue deformations that are not constrained by the force balance of cortical tensions. This means that weaker contributions to stress, e.g., from intermediate filaments, microtubules, and the nucleus in the cell interior, as well as externally applied stresses will drive the isogonal modes. This has two important consequences: (*i*) For the analysis of experimental data, it implies that we can decompose the tissue deformations into active, tension-driven, and passive, isogonal contributions. (*ii*) For modeling, it implies that we need to specify a cell shape elasticity to fix the isogonal modes.

In the following, we discuss the details of the *tension–isogonal decomposition*. To achieve a well-defined decomposition of tissue deformation, we need to specify a reference tessellation that is uniquely determined by the tension triangulation. There will then be a unique isogonal deformation that deforms the reference configuration into the physical cell array. Fortunately, we do not need to explicitly construct the reference configuration. Rather, we use the triangulation formed by the cell centroids to quantify the physical tissue shape (*Merkel et al., 2017*). (This implies to some degree of coarse-graining, as information about the non-affine displacements of individual vertices is lost). We can then obtain the isogonal deformation by comparing the tension triangulation to the centroidal triangulation.

### Single-vertex level

To make this construction concrete, we first consider a single tri-cellular vertex, with the associated tension triangle $\hat{\mathbf{T}}_{ij}$ and the centroids $\mathbf{c}_i$. The indices $i, j = 1, 2, 3$ label the three cells that meet at the vertex (see *Figure 6—figure supplement 1A'*). The tension triangle vectors $\hat{\mathbf{T}}_{ij}$ are obtained by local tension inference based on the angles at the vertex (see Appendix 2). The orientation of the tension vectors is determined by demanding that they are at right angles to the cell-cell interfaces $\mathbf{e}_{ij}$. To fix the tension scale, we normalize the area of the tension triangle to unity. Fixing the tension triangle area is justified because it maintains the total area of the tension triangulation such that cell dilation/contraction is purely isogonal. The numerical value of the tension triangle area is immaterial since it can be absorbed in the scale factor $\ell_0$.

We define the local isogonal deformation tensor $\mathbf{I}_{\text{vertex}}$ such that it transforms the tension triangle into the centroidal triangle. Mathematically, this definition is written as the equations $\ell_0 \, \mathbf{I}_{\text{vertex}}.\hat{\mathbf{T}}_{ij} = \mathbf{c}_j - \mathbf{c}_i, \ i \neq j \in \{1, 2, 3\}$. Since there are two independent displacement vectors in tension space and physical space each, these equations represent four independent constraints that uniquely determine the four components of the $2{\times}2$ matrix $\ell_0 \mathbf{I}_{\text{vertex}}$. The scale factor $\ell_0$ converts units of tension (a.u.) into units of length. How we fix $\ell_0$ is described below.

The isogonal deformation tensor $\mathbf{I}_{\text{vertex}}$ can be decomposed into a symmetric tensor $\mathbf{U}$ and a rotation $\mathbf{R}$ using polar decomposition $\mathbf{UR} = \mathbf{I}_{\text{vertex}}$. The appearance of a rotational component in the isogonal deformation might appear surprising at first glance since isogonal modes leave angles at vertices invariant. However, the centroidal triangle can be sheared relative to the tension triangle by a simple shear – composition of pure shear with a rotation – while the angles at the vertex remain invariant.

#### Fixing the scale factor $\ell_0$

To fix $\ell_0$ we use that the tension triangles are initially close to equilateral (as evidenced by the angle distribution peaked around $\pi/3$, see *Figure 8L*). Recall that we have normalized the area of the tension triangle to unity by convention. An equilateral tension triangle with a unit area has an edge

length $2/3^{1/4}$. To find $\ell_0$, we compare this value to the average edge length of the *centroidal triangles*, i.e., the average centroid distance between adjacent cells, which we find to be $6.38\,\mu$m. The ratio of these two numbers gives a scale factor $\ell_0 \approx 4.2\,\mu$m. Note that the scale factor $\ell_0 \approx 4.2\,\mu$m minimizes the isogonal strain at $t = 0$ (see *Figure 6—figure supplement 1B*), indicating that at early times, the cell array geometry closely matches Voronoi dual of the tension triangulation. Indeed, we can also calculate $\ell_0$ from the initial average cell-cell interface length $\langle\ell(0)\rangle \approx 3.6\,\mu$m. To this end, we use that the Voronoi dual to an equilateral triangulation with edge length $T = 2/3^{1/4}$ is a hexagonal lattice with edge length $T/\sqrt{3} \approx 0.88$, giving $\ell_0 \approx 4.1\,\mu$m. Finally, minimizing $\langle\|\mathbf{I}\|\rangle$ at a reference time provides an alternative way to determine $\ell_0$ in cases where the initial triangulation is not as uniform and regular as in the *Drosophila* blastoderm.

## Coarse-grained isogonal strain

To find the tissue scale isogonal strain, we coarse grain the isogonal deformation tensor on a square grid in the flat map projection ('pullback'). The grid spacing is set to $20\,\mu$m, such that each grid facet contains ca 10–15 cells (grid facets near the poles contain fewer cells because of the distortion caused by the pullback projection). We calculate $\mathbf{I}_{\text{vertex}}$ for all tri-cellular vertices in a grid facet and take the average to find $\mathbf{I}_{\text{facet}}$.

Figures *Figure 4D* and *Figure 6—figure supplement 1B–D* (top panels) show the isogonal strain tensor $\mathbf{I}_{\text{facet}} - \mathbb{I}$, where   is the identity matrix. The bottom panels in *Figure 6—figure supplement 1B–D* show the individual, single-vertex strain tensors $\mathbf{I}_{\text{facet}} - \mathbb{I}$ at each vertex. Crosses indicate the principal eigenvectors of the strain tensors. The bar length is proportional to the eigenvalue. Green (magenta) indicates a positive (negative) eigenvalue, i.e., contraction (elongation).

To calculate the average isogonal deformation in different tissue regions (*Figure 4E*), we first transform the single-vertex isogonal deformation tensor to the local co-rotating ('Lagrangian') frame (see Appendix 1.1). This is important because the germ band rotates as it moves over the posterior pole during elongation.

To estimate the critical tension for T1s in the lateral ectoderm, we need the average isogonal contribution to the length of DV-oriented interfaces, $\langle\Delta\ell_{\text{iso}}\rangle$ (see Appendix 3). At the end of VF invagination ($t = 25$ min), we find $\langle(\mathbf{I}_{\text{vertex}})_{22}\rangle \approx 1.538 \pm 0.006$, where the index 2 denotes the DV component and the average is taken over cells in the lateral ectoderm (region shaded in blue in *Figure 6—figure supplement 1C*). We thus have $\langle\Delta\ell_{\text{iso}}\rangle = \langle(\mathbf{I}_{\text{vertex}})_{22} - 1\rangle\langle\ell(0)\rangle \approx 1.9\,\mu$m for DV-oriented interfaces, where $\langle\ell(0)\rangle \approx 3.5\,\mu$m is the average initial interface length.

*Figure 7—figure supplement 1* shows the coarse-grained isogonal strain in a *snail* mutant, where the ventral furrow is abolished. Without the ventral furrow invagination pulling on the lateral ectoderm, no isogonal strain is found in the lateral ectoderm before and during germ-band extension. This validates that isogonal strain reliably indicates tissue deformation driven by external forces, and, vice versa, the absence of isogonal strain indicates the lack of external driving forces.

## Filtering of degenerate tension triangles

When one of the angles at a vertex is close to $\pi$, the tension triangle becomes highly elongated. In fact, sometimes one of the angles at a vertex exceeds $\pi$ due to fluctuations and inaccuracies in detection. In the tension inference, this leads to negative tensions. The (nearly) degenerate tension triangles, in turn, lead to extreme local isogonal deformations. We, therefore, exclude nearly degenerate tension triangles whose perimeter is larger than 8 after normalizing the area to 1 (ca. 2% of the tension triangles are filtered out this way).

## Local tension anisotropy

From the tension vectors $\mathbf{T}_{ij}$ at a vertex, we calculate the 'tension anisotropy' tensor

$$\mathbb{T} = \frac{1}{\mathcal{N}^2} \sum_{(ij)} \mathbf{T}_{ij} \otimes \mathbf{T}_{ij} \tag{10}$$

where the sum runs over pairs from the set of the three cells that meet the vertex. The normalization factor $\mathcal{N}^2 = \sum_{(ij)} T_{ij}^2$ is chosen such that $\text{Tr}\,\mathbb{T} = 1$. The difference of the eigenvalues of $\mathbb{T}$ quantifies the magnitude of tension anisotropy. When the interface angles at the vertex are all identical to $120°$ the tension anisotropy vanishes. For non-identical vertex angles the tension is locally anisotropic and the dominant eigenvector of $\mathbb{T}$ points along the principal axis of stress. Anisotropic tension

also implies that the tension triangle is non-equilateral. The tension triangle's axis of elongation is perpendicular to the principal axis of tension because the tension triangle vectors are rotated by 90° relative to the interfaces (*Figure 2B*).

In passing, we note that the tension anisotropy tensor is closely related, but not exactly equal, to the stress tensor $\sigma$. Instead, it has the form of a metric tensor and thus describes the local geometry of tension space. Specifically, the $\mathbb{T}$ measures the *extrinsic* anisotropy of tension and is complementary to the *intrinsic* shape tensor that we will introduce in the next section.

## Tension-isogonal decomposition for cell quartets

The calculation for cell quartets (vertex pairs) is analogous to the case of single vertices. The two tension triangles associated with the inner vertices of the cell quartet form a 'kite' (see *Figure 4— figure supplement 1A*). We normalize tension such that the average area of the tension triangles is one. The quartet's isogonal deformation tensor $\mathbf{I}_{\text{quartet}}$ is defined such that it transforms the vectors $\mathbf{T}_{ij}$ into the corresponding centroidal displacement vectors $\mathbf{c}_j - \mathbf{c}_i$. Here, we only use the vectors around the perimeter of the tension and centroidal kite. This ensures that the isogonal deformation tensor varies continuously through the cell neighbor exchange where the inner edge in the kite is flipped. As in the single-vertex case, we find $\mathbf{I}_{\text{quartet}}$ using least squares.

To quantify the deformations of the tension kite and the physical cell quartet, we define the shape tensors

$$\mathbb{G}_{\text{T}} = \ell_0^2 \sum_{(ij)} \hat{\mathbf{T}}_{ij} \otimes \hat{\mathbf{T}}_{ij}, \tag{11}$$

$$\mathbb{G}_{\text{C}} = \sum_{(ij)} (\mathbf{c}_i - \mathbf{c}_j) \otimes (\mathbf{c}_i - \mathbf{c}_j), \tag{12}$$

where the index pairs $(ij)$ run over cell pairs going around the quartet, i.e., $(ij) = (12), (23), (34), (41)$. Notably, with these definitions, one has the relation

$$\mathbf{I}_{\text{quartet}}.\mathbb{G}_{\text{T}}.\mathbf{I}_{\text{quartet}} \approx \mathbb{G}_{\text{C}}. \tag{13}$$

## Remarks on the Voronoi construction for the reference cell array

For the tension–isogonal decomposition, we have circumvented the explicit construction of a reference cell array by directly comparing the tension triangulation to the centroidal triangulation of the physical cells. This approach is particularly useful because a local isogonal deformation tensor can be obtained on the level of a single tri-cellular vertex.

The explicit construction of the reference cell array allows us to understand the geometry of cell-neighbor exchanges and find a criterion for when they occur (see main text for a discussion of a simple symmetric case and the companion paper [*Claussen et al., 2024*] for the general case). We define the reference tesselation using a generalized Voronoi construction based on the tension triangulation. The vertices of this generalized Voronoi tessellation are given by the circumcircle centers of the triangles in the triangulation. The generalized Voronoi tessellation is only free of self-intersections when the triangulation fulfills the Delaunay criterion that opposite angles in adjacent triangles must sum to a value smaller than $\pi$. As we will see, in the following section, this provides a purely geometric criterion for neighbor exchanges in the absence of isogonal strain. In the presence of isogonal strain, the reference can have self-intersections (i.e. negative interface lengths). This is similar to how the reference state encoding plastic deformations in finite strain theory isn't necessarily geometrically compatible (*Efrati et al., 2009*).

In general, the centroids of the generalized Voronoi cells will not coincide with the vertices of the tension triangulation. This means that the definition of the isogonal deformation tensor based on the tension triangulation vertices is not exactly identical to that based on an explicit Voronoi construction. However, the two agree for a periodic lattice of identical tension triangles. In this case, the vertices of the tension triangulation coincide exactly with the centroids of the generalized Voronoi tessellation. In other words, using the tension triangulation vertices to define isogonal deformations amounts to approximating tissue as locally lattice-like, which is valid when spatial gradients are small on the cell scale.

In passing, we note that the generalized Voronoi tessellation is not the only possible reference state. Any tessellation that is uniquely defined by a given tension triangulation is in principle a

valid reference state. The Voronoi tessellation is a useful choice for the following analysis because the Voronoi cell shapes behave similarly to the actual cell shapes and because it is defined purely geometrically. Other choices of the reference state might be of use depending on the particular question and when one has more detailed information about the mechanical properties of the cells.

## Isognal potential and isogonal pure shear

As discussed in the main text (and further elaborated in *Claussen et al., 2024*) our simulations show that a non-zero cell-level shear modulus is required for extension via active T1s. Here, we analyze the relationship of cell-level and tissue-level shear modulus. Crucially, because of the dominance of cortical tensions, a tissue patch in our model will respond to externally applied forces via an isogonal deformation. We first show that an isogonal deformation can create tissue-scale pure shear (on the level of cell centroid displacement – the transformation of cell vertices is necessarily non-affine to preserve angles). Then we show that these pure shear modes correspond to the response of the tissue to external force by computing the Hessian of our cell elastic energy in the subspace spanned by isogonal deformations, and measure the shear modulus.

To show that isogonal deformations can shear and not just dilate/contract cells, we make use of the'isogonal mode parametrization introduced in *Noll et al., 2017*. It assigns an isogonal 'potential' $\Theta_i$ to each cell, and calculates the cell displacements from the $\Theta_i$ and the edge tension vectors. In the following, we will show that a constant gradient in the isogonal potential generates a uniform translation in real space. By integration, this implies that a quadratic spatial profile of the isogonal potential creates a pure shear.

Let us identify the real space edge unit vectors by the two adjacent cells $\mathbf{e}_{ij} = -\mathbf{e}_{ji}$ and denote the corresponding tensions as $T_{ij}$. We denote by $\hat{\mathbf{a}}$ the vector $\mathbf{a}$ rotated by $\pi/2$ in counterclockwise direction, such that $\hat{\mathbf{a}}$ fulfills the relations $\hat{\mathbf{a}}.\mathbf{a} = 0$ and $\|\hat{\mathbf{a}}\| = \|\mathbf{a}\|$. In force balance, the rotated force vectors $\hat{\mathbf{T}}_{ij} = T_{ij}\hat{\mathbf{e}}_{ij}$ form a triangulation (*Noll et al., 2017*).

The isogonal displacement $\mathbf{u}_{ijk}$ of the real space vertices $\mathbf{r}_{ijk}$ (identified by the three adjacent cells) is given by

$$\mathbf{r}_{ijk} \rightarrow \mathbf{r}_{ijk} + \mathbf{u}_{ijk} = \mathbf{r}_{ijk} + \frac{1}{S_{ijk}}\left[\Theta_i\mathbf{T}_{jk} + (\text{cyc.})\right] \tag{14}$$

where $S_{ijk} = \hat{\mathbf{T}}_{ij}.\mathbf{T}_{ik}$ is the area of the tension triangle $(ijk)$ and cyc. denotes cyclic permutations of $(ijk)$.

First, observe that the uniform isogonal mode $\Theta_i = $ const. has no effect on the vertex positions because

$$\mathbf{T}_{jk} + \mathbf{T}_{ki} + \mathbf{T}_{jk} = 0 \quad (\text{force balance}). \tag{15}$$

Now we aim to show that a constant gradient in $\Theta_i$ drives a uniform displacement of the $\mathbf{r}_{ijk}$. Specifically, by uniform gradient, we mean $\Theta_i = \mathbf{t}_i.\mathbf{a}$, i.e., a linear gradient in the tension space ($\mathbf{t}_i$ is the position of the tension triangulation vertex corresponding to cell $i$, such that $\hat{\mathbf{T}}_{ij} = \mathbf{t}_j - \mathbf{t}_i$). To show that the displacement in real space is uniform, it is enough to show that two adjacent vertices are displaced identically. By induction, this implies that all displacements are identical. It is, therefore, sufficient to consider a quartet of cells ($i = 1-4$), corresponding to a 'kite' in tension space (note that $\hat{\mathbf{a}}.\mathbf{b}$ is identical to the wedge product $\mathbf{a} \wedge \mathbf{b}$).

Because a constant can be arbitrarily added to all $\Theta_i$, we can set $\Theta_1 = 0$ and thus have $\Theta_i = \hat{\mathbf{T}}_{ij}.\mathbf{a}$ for $i = 2, 3, 4$. The displacements now read

$$\mathbf{u}_{123} = \frac{1}{S_{123}}\left(\hat{\mathbf{T}}_{12}.\mathbf{a}\,\mathbf{T}_{31} + \hat{\mathbf{T}}_{13}.\mathbf{a}\,\mathbf{T}_{12}\right) \tag{16}$$

$$\mathbf{u}_{134} = \frac{1}{S_{134}}\left(\hat{\mathbf{T}}_{13}.\mathbf{a}\,\mathbf{T}_{41} + \hat{\mathbf{T}}_{14}.\mathbf{a}\,\mathbf{T}_{13}\right) \tag{17}$$

To show that these displacements are identical, we project them onto two conveniently chosen, linearly independent vectors, namely $\hat{\mathbf{T}}_{12}$ and $\hat{\mathbf{T}}_{13}$. For the latter, we find

$$\hat{\mathbf{T}}_{13}.\mathbf{u}_{123} = \frac{1}{S_{123}}\hat{\mathbf{T}}_{13}.\mathbf{a}\,\hat{\mathbf{T}}_{13}.\mathbf{T}_{12} = -\hat{\mathbf{T}}_{13}.\mathbf{a}, \tag{18}$$

$$\hat{\mathbf{T}}_{13}.\mathbf{u}_{134} = \frac{1}{S_{134}} \hat{\mathbf{T}}_{13}.\mathbf{a} \ \hat{\mathbf{T}}_{13}.\mathbf{T}_{14} = -\hat{\mathbf{T}}_{13}.\mathbf{a}, \tag{19}$$

where we used that $\hat{\mathbf{T}}_{ij}.\mathbf{T}_{ij} = 0$ and applied the definition of $S_{ijk}$.

Projecting (*Equation 20*) and (*Equation 21*) onto $\hat{\mathbf{T}}_{12}$ gives

$$\hat{\mathbf{T}}_{12}.\mathbf{u}_{123} = \frac{1}{S_{123}} \hat{\mathbf{T}}_{12}.\mathbf{a} \ \hat{\mathbf{T}}_{12}.\mathbf{T}_{31} = -\hat{\mathbf{T}}_{12}.\mathbf{a} \tag{20}$$

$$\hat{\mathbf{T}}_{12}.\mathbf{u}_{134} = \frac{1}{S_{134}} \left( \hat{\mathbf{T}}_{13}.\mathbf{a} \ \hat{\mathbf{T}}_{12}.\mathbf{T}_{41} + \hat{\mathbf{T}}_{14}.\mathbf{a} \ \hat{\mathbf{T}}_{12}.\mathbf{T}_{13} \right) \tag{21}$$

To show the equality of these two right-hand sides, we use that given $\hat{\mathbf{T}}_{13}.\mathbf{a}$ and $\hat{\mathbf{T}}_{14}.\mathbf{a}$ we can find $\mathbf{a}$ and substitute the result into $\hat{\mathbf{T}}_{12}.\mathbf{a}$. We start by 'expanding the identity'

$$\begin{pmatrix} -\hat{\mathbf{T}}_{13}- \\ -\hat{\mathbf{T}}_{14}- \end{pmatrix} \mathbf{a} = \begin{pmatrix} \hat{\mathbf{T}}_{13}.\mathbf{a} \\ \hat{\mathbf{T}}_{14}.\mathbf{a} \end{pmatrix} \quad \Rightarrow \quad \mathbf{a} = \begin{pmatrix} -\hat{\mathbf{T}}_{13}- \\ -\hat{\mathbf{T}}_{14}- \end{pmatrix}^{-1} \begin{pmatrix} \hat{\mathbf{T}}_{13}.\mathbf{a} \\ \hat{\mathbf{T}}_{14}.\mathbf{a} \end{pmatrix} \tag{22}$$

Explicitly writing out the inverse matrix then gives

$$\mathbf{a} = \frac{1}{\hat{\mathbf{T}}_{13}.\mathbf{T}_{14}} \begin{pmatrix} | & | \\ \mathbf{T}_{14} & -\mathbf{T}_{13} \\ | & | \end{pmatrix} \begin{pmatrix} \hat{\mathbf{T}}_{13}.\mathbf{a} \\ \hat{\mathbf{T}}_{14}.\mathbf{a} \end{pmatrix} \tag{23}$$

With this, we find the relation

$$\hat{\mathbf{T}}_{12}.\mathbf{a} = -\frac{1}{S_{134}} \left( \hat{\mathbf{T}}_{12}.\mathbf{T}_{41} \ \hat{\mathbf{T}}_{13}.\mathbf{a} + \hat{\mathbf{T}}_{12}.\mathbf{T}_{13} \ \hat{\mathbf{T}}_{14}.\mathbf{a} \right) \tag{24}$$

where we used $\mathbf{T}_{ij} = -\mathbf{T}_{jk}$ to flip the indices on $\mathbf{T}_{14}$. Comparing to (*Equation 24*) and (*Equation 25*) now shows the identity of their RHSs.

Taken together, we have shown that

$$\hat{\mathbf{T}}_{12}.\mathbf{u}_{123} = \hat{\mathbf{T}}_{12}.\mathbf{u}_{134} \quad \text{and} \quad \hat{\mathbf{T}}_{13}.\mathbf{u}_{123} = \hat{\mathbf{T}}_{13}.\mathbf{u}_{134}, \tag{25}$$

Because $\hat{\mathbf{T}}_{12}$ and $\hat{\mathbf{T}}_{13}$ are linearly independent, it follows that $\mathbf{u}_{123} = \mathbf{u}_{134}$. QED.

We just showed that a constant gradient in $\Theta_i$ corresponds to a uniform displacement of the real space vertices $\mathbf{r}_{ijk}$. We can, therefore, think of $\Theta(\mathbf{t})$ as a 'potential' for the isogonal displacement field: $\mathbf{u} \approx \nabla_\mathbf{t}\Theta$, where the approximation is valid for slowly varying gradients and exact for constant gradients. The gradient $\nabla_\mathbf{t}$ is taken in *tension space* because the function $\Theta(\mathbf{t}_i) = \Theta_i$ is defined on the vertices of the tension triangulation $\mathbf{t}_i$.

A pure shear aligned with the coordinate axes is given by a displacement field $\mathbf{u}(\mathbf{r}) = \varepsilon \, \text{diag}\,(1, -1).\mathbf{r}$ and is, therefore, generated (approximately) by a isogonal potential forming a hyperbolic paraboloid $\Theta = \varepsilon \, \mathbf{t}^\mathrm{T}.\text{diag}\,(-1, 1).\mathbf{t} = \varepsilon \left( t_1^2 - t_2^2 \right)$.

## Appendix 4

### Local tension configurations

In the main text, we argued that the local configuration of tensions at a vertex is characterized by the shape of the tension triangle. Specifically, acute triangles correspond to tension cables, while obtuse triangles correspond to tension bridges (see *Figure 8—figure supplement 1A*).

Whether a triangle is obtuse or acute is an *intrinsic* property, i.e., it is independent of the position, orientation, and size of the triangle in the embedding space. The intrinsic shape of a triangle is given by its angles. Since these angles sum to $\pi$, we can represent the space of triangle shapes in barycentric coordinates (see *Figure 8—figure supplement 1B*). The center of the barycentric shape space corresponds to an equilateral triangle, while the edges correspond to triangles where one angle is zero and the corners to triangles where two angles are zero. Because the order of labeling the angles in the triangle is arbitrary, the shape space is invariant under permutations of the angles and a single fundamental domain (highlighted in *Figure 8—figure supplement 1B*) is representative.

In the triangle shape space, we have colored each point according to how anisotropic and how acute-vs-obtuse it is. We already know that we can read off the magnitude of anisotropy from the eigenvalues of the tensor $\mathbb{T}$, defined in the previous subsection. To quantify the shape (acute vs obtuse), we define a second tensor

$$(\mathcal{S})_{IJ} = \frac{1}{\mathcal{N}^2}\, \mathbf{T}_I \cdot \mathbf{T}_J, \tag{26}$$

where the indices $I, J \in \{1, 2, 3\}$ are short-hand labels for the three interfaces that meet a tri-cellular vertex. Because $\sum_I \mathbf{T}_J = 0$ (force balance), this tensor has the null vector $(1, 1, 1)^{\mathrm{T}}$. To get rid of this nullspace, we go to barycentric coordinates by defining the pair of vectors, $\boldsymbol{\tau}_{1,2}$ that we pack into a matrix

$$\mathfrak{T} = \begin{pmatrix} \boldsymbol{\tau}_1 \\ \boldsymbol{\tau}_2 \end{pmatrix} = \frac{1}{\sqrt{2}\mathcal{N}} \begin{pmatrix} \mathbf{T}_1 - \mathbf{T}_2 \\ \mathbf{T}_3/\sqrt{3} \end{pmatrix}. \tag{27}$$

These vectors have the defining property that they are orthogonal to one another and to the null vector $(1, 1, 1)^{\mathrm{T}}$ of $\mathcal{S}$. Now we can define the shape tensor

$$\mathbb{S} = \mathfrak{T}\mathfrak{T}^{\mathrm{T}} = \begin{pmatrix} \|\boldsymbol{\tau}_1\|^2 & \boldsymbol{\tau}_1 \cdot \boldsymbol{\tau}_2 \\ \boldsymbol{\tau}_1 \cdot \boldsymbol{\tau}_2 & \|\boldsymbol{\tau}_2\|^2 \end{pmatrix}. \tag{28}$$

Observe that

$$\mathfrak{T}^{\mathrm{T}}\mathfrak{T} = \boldsymbol{\tau}_1 \otimes \boldsymbol{\tau}_1 + \boldsymbol{\tau}_2 \otimes \boldsymbol{\tau}_2 = \mathbb{T}, \tag{29}$$

which implies that $\mathbb{T}$ and $\mathbb{S}$ share the same eigenvalues. In fact, we can perform a singular value decomposition of $\mathfrak{T}$

$$\mathfrak{T} = R(\psi/2)\boldsymbol{\Lambda}R^{\mathrm{T}}(\phi), \tag{30}$$

where $R(\phi)$ is the rotation matrix with angle $\phi$ and $\boldsymbol{\Lambda} = \mathrm{diag}\,(\lambda_1, \lambda_2)$ is a diagonal matrix of singular values whose squares $\lambda_1^2, \lambda_2^2$ are the eigenvalues of $\mathbb{T}$ and $\mathbb{S}$. Because we normalized the tension vectors, $\mathrm{Tr}\,\mathbb{T} = \mathrm{Tr}\,\mathbb{S} = \lambda_1^2 + \lambda_2^2 = 1$ is the angle of the dominant eigenvector in physical space, which gives the orientation of the principal axis of stress. As we will see below, the angle $\psi$ indicates whether the triangle is obtuse or acute. The reasoning to define the rotation based on $\psi/2$ will become apparent below when we represent the shape tensor by a complex number.

The shape tensor $\mathbb{S}$ has only two degrees of freedom, the difference of eigenvalues $\lambda_1^2 - \lambda_2^2$ (which it shares with the tensor $\mathbb{T}$) and the angle $\psi$ that sets the orientation of its two orthogonal eigenvectors. We can represent $\mathbb{S}$ by a complex number

$$\Psi = \frac{\tau_1^2 - \tau_2^2 + 2i\,\boldsymbol{\tau}_1 \cdot \boldsymbol{\tau}_2}{\tau_1^2 + \tau_2^2} = |\Psi|e^{i\psi}, \tag{31}$$

where $|\Psi| = |\lambda_1^2 - \lambda_2^2|$ is the magnitude of tension anisotropy and the phase $\psi$ is the 'shape space angle' defined by *Equation 30*.

Permutations of the edge indices $\alpha$ correspond to sign changes and rotations of $\psi$ by integer multiples of $2\pi/3$. To mod out these group actions (which correspond to the symmetries of the barycentric shape space *Figure 8—figure supplement 1B*), we define

$$\tilde{\psi} = |3\psi \bmod_\pi 2\pi| \in [0, \pi], \tag{32}$$

where $x \bmod_d n = (x + d \bmod n) - d \in [-d, n - d]$ is the modulo operation with offset $d$. This shape parameter informs how acute or obtuse a tension triangle is. For an acute isosceles triangle, $\tilde{\psi} = 0$, whereas for an obtuse isosceles triangle $\tilde{\psi} = \pi$. For non-isosceles triangles, $\tilde{\psi}$ takes values in $[0, \pi]$ quantifying the relative degree of acuteness vs obtuseness. For an equilateral triangle, $|\Psi| = 0$ and $\psi$ is not defined. The axes $|\Psi|$ and $\tilde{\psi}/\pi$ span the space of triangle shapes shown in *Figure 8—figure supplement 1*.

As an alternative to the above mathematical derivation of the shape parameter, we briefly discuss a more intuitive parameter as follows: For a given tension triangle, label its edges in ascending order of length (i.e. $T_2$ and $T_3$ will be second highest and highest tension, respectively). We can now define a parameter

$$p = 2\frac{|T_2 - T_3|}{T_2 + T_3} \in [0, 1], \tag{33}$$

which quantifies the relative tension difference between the two largest tensions at a vertex. For a tension cable, there are two large tensions that are similar in magnitude, implying that $p$ is small. For a tension bridge, there is one high-tension edge meeting two low-tension edges so $p$ will be closer to 1. The maximal value $p = 1$ is reached for $T_1 = T_2 = T_3/2$. Remarkably, to a very good approximation $p \approx |\Psi|\tilde{\psi}/\pi$.

## Random Delaunay triangulation

As a reference for the triangle shape distributions, we generated random Delaunay triangulations based on the Ginibre random point process (*Ginibre, 1965*; the Ginibre process models the distribution of particles with repulsion and thus generates slightly more regular point distribution than the Poisson process where points are randomly positioned independently from one another). A small sample of such a Ginibre-based random triangulation is shown in *Figure 8—figure supplement 2A*. The histograms in the triangle shape space *Figure 8—figure supplement 2B* show good agreement between the Ginibre-based random Delaunay triangulation and the distribution of tension triangle shapes at the end of GBE. Moreover, the marginalized angle distribution (black dashed line), closely matches the experimentally observed distribution (*Figure 8—figure supplement 2D*).

These findings suggest that the local coordination of cortical tensions is lost during GBE and that the local tension configurations approach a maximally disordered state.

## Appendix 5

### Simulation methods

In the following, we explain the simulation details for **Figure 3**, where we present a model of a single intercalating quartet. The code for the simulations shown is available online: https://github.com/nikolas-claussen/geometric_basis_of_convergent_extension_simulation (copy archived at **Claussen, 2024b**).

Our model is based on two assumptions: adiabatic force balance, and dominance of cortical tensions. The first assumption means that we obtain the instantaneous cell geometry by solving force-balance equations. The second assumption means that we do so in two steps: first, we determine the interface angles from the cortical tension force balance. The remaining forces in the system are much weaker than cortical tensions and only affect the residual degrees of freedom, the isogonal modes.

In this work, we only consider a symmetric lattice of identical tension triangles, so several simplifications occur (the general case of a disordered cell array is the subject of the companion paper **Claussen et al., 2024**). In this case, the cell shape is completely specified by the three vertex angles $\phi_i$ and edge lengths $\ell_i$ at each vertex, with $i \in \{0, 1, 2\}$. Because the cell array is a symmetric lattice, the edge lengths $\ell_i$ can be changed without affecting the angles and, therefore, parameterize the isogonal degrees of freedom.

We assume that the dynamics of the tensions are determined by a mechano-sensitive feedback loop implementing positive tension feedback. To account for the dynamics of a junction post T1, each junction at a vertex is characterized by the total tensions $T_i$, and the passive tension $T_{p,i}$. The tensions at a vertex evolve according to:

$$\dot{T}_i = \tau_{\mathrm{T}}^{-1} T_i^n - \tau_p^{-1} T_{p,i} - \frac{1}{3} \sum_k \left( \tau_{\mathrm{T}}^{-1} T_k^n - \tau_p^{-1} T_{p,k} \right) \tag{34}$$

where $n > 1$ and $\tau_{\mathrm{T}}$ is a time scale converting simulation time into minutes (fit to the data). The passive tension is 0 on all junctions except those that were newly created by a T1 transition. The initial value of $T_p$ on those interfaces is discussed below. Given the tensions, we first calculate the angles from the tensions using the law of sines as $\phi_i = \sin^{-1}\left(\frac{T_i}{2R}\right)$, where $R$ is the tension triangle circumradius.

### Cell shape energy

To determine the $\ell_i$, we minimize an elastic energy based on the cell shape tensor:

$$S_{\mathcal{C}} = \sum_i \ell_i \mathbf{e}_i \otimes \mathbf{e}_i \tag{35}$$

where $i$ runs over the edges of the cell $\mathcal{C}$, and $\mathbf{e}_i$ are the unit vectors pointing along the interfaces. Note that in this shape tensor, each edge contributes linearly in length to the cell shape. This means that artificially subdividing an edge has no effect on the cell shape tensor. This makes sense if we assume that the elasticity we aim to model using $S$ resides in the cell interior (cytoplasm incompressibility, microtubules, intermediate filaments, nucleus). The shape tensor can also be defined using vectors from the cell centroid to its vertices (in the lattice, the two definitions are equivalent). An alternative definition of the elastic energy using $\tilde{S}_{\mathcal{C}} = \sum_i \ell_i^2 \mathbf{e}_i \otimes \mathbf{e}_i$ instead gives broadly similar results (although interface collapse happens more abruptly, because of the higher order non-linearity).

We assign a reference shape $S_0$. An isotropic $S_0 \propto \mathrm{Id}$ favors equilateral hexagons. We chose a slightly anisotropic reference shape $S_0$ (15% anisotropy) to model the isogonal stretching caused by the ventral furrow before the onset of GBE. This anisotropy sets the angle at which the inner interface collapses, $\ell_1 = 0$. The experimental value of the collapse tension was, therefore, used to fit the $S_0$-anisotropy.

We define the cell's elastic energy via

$$E_{\mathcal{C}} = \lambda [\mathrm{Tr}(S_{\mathcal{C}} - S_0)]^2 + \mu \mathrm{Tr}[(S_{\mathcal{C}} - S_0)^2] \tag{36}$$

with bulk modulus $\lambda$ and shear modulus μ. We used a shear/bulk ratio of $\mu/\lambda = 0.1$. Because of the separation of scales between cortical tensions and elastic energy baked into the model, the absolute

values of $\lambda, \mu$ are irrelevant. Furthermore, for the case of active T1s, the results do not depend on $\mu/\lambda$, as long as $\mu > 0$.

The edge lengths $\ell_i$ can now be determined by minimizing the elastic energy *Equation 36* w.r.t. the $\ell_i$, for the angles determined by the tension dynamics.

## Comparison to area-perimeter elastic energy

Strikingly, within the single-quartet model, the 'shape strain' $S_\mathcal{C} - S_0$ can always be set to 0 by choice of the edge lengths, and the energy $E_\mathcal{C} = 0$ throughout. This can already be seen from a degree-of-freedom count (three $\ell_i$ for the three independent components of $S_\mathcal{C} - S_0$). The elastic energy, therefore, acts only on the isogonal modes (i.e. $\partial_{\phi_i} E_\mathcal{C}|_{\ell_i=\text{minimizers}} = 0$) and there is no energy barrier for intercalations. Consequently, there is no need for noise to drive intercalations in our model. By contrast, for the widely-used area-perimeter elastic energy $E = (A - A_0)^2 + (P - P_0)^2$ (where $A, P$ are the cell area and perimeter, and $A_0, P_0$ their target values) (*Bi et al., 2015*), there exists an energy barrier, and the inner interface $\ell_1$ only collapses when $\phi_0 = \pi$. This is shown in *Figure 3—figure supplement 1B*. Note that the area-perimeter energy is a special case because of the geometric incompatibility of area and perimeter constraints. Combining area elasticity with shear elasticity based on the shape tensor, $E = (A - A_0)^2 + \mu\text{Tr}[(S_\mathcal{C} - S_0)^2]$, (or perimeter elasticity with cell shape bulk elasticity) leads to similar results as *Equation 36*. Note also that because of the degree-of-freedom count, the system is under-specified if the shear modulus is 0, foreshadowing the fact that without shear modulus, no convergence-extension takes place (*Claussen et al., 2024*).

## Myosin handover and passive tension

When an interface collapses, $\ell_i = 0$, the tension triangulation is modified topologically: the edge corresponding to the collapsed interfaces is replaced by one corresponding to the new interface (triangulation 'flip'). To complete our model, we must specify the initial conditions of this new edge.

As illustrated in *Figure 3D*, we propose a myosin handover mechanism to explain the extension of the new interface post-intercalation. An interface with cortical tension $T$ is comprised of the adherens-junctional actomyosin cortex of the two adjacent cells, which are coupled mechanically via adherens junctions. Under force balance, the total tension $T$ has to be constant along the cortex, but the individual tensions on either side can be non-uniform, as the resulting traction forces are exchanged via adherens junctions. In the following, we assume as a first-order approximation that the level of active tension (i.e. myosin concentration) varies linearly along an interface (similar calculations have been performed in *Kale et al., 2018* to calculate interfacial shear stress). This allows us to geometrically obtain the myosin concentration at the individual cortices that will form the two juxtaposed sides of the new interface (see Fig *Figure 3D*).

Consider the tensions $T_0, T_1, T_2$ at a vertex, and let 0 be the interface about to collapse. The three cells that meet at the vertex will be referred to by $(01), (12), (21)$ (where e.g. $(01)$ is the cell abutting interfaces 0 and 1). Let $m_{01}, m_{12}, m_{21}$ be the motor molecule concentrations (in units of tension) at the vertex in the junctional cortices of the three cells. Then, the tensions are related to the motor molecule concentrations:

$$T_0 = m_{21} + m_{01}, \quad T_1 = m_{01} + m_{12}, \quad T_2 = m_{12} + m_{21}.$$

This uses the assumption of myosin continuity at vertices and the fact that the tension on an interface is the sum of the tensions of the two cortices that make it up. The motor molecule concentration on the cortex belonging to the new interfaces, post-collapse, will be equal to $m_{12}$. Solving for this in terms of the tensions:

$$m_{12} = \frac{T_1 + T_2 - T_0}{2}$$

The new interface consists of two cortices, coming from the two vertices of the old junction. Let the tensions at the two triangles be $T_0, T_1, T_2$ and $T_0, T_1', T_2'$. Let $T_a$ be the active tension on the new junction immediately after the T1. It is equal to

$$T_a = \frac{(T_1 + T_2 - T_0) + (T_1' + T_2' - T_0)}{2} \tag{37}$$

Note, however, that the total tension $T_n$ on the new junction is not necessarily equal to $T_a$. The total tension is defined geometrically from the angles at the new junction (or, equivalently, the tension triangle vertices). Indeed, generally, $T_n > T_a$, i.e., the active tension on the new junction is not enough to balance the tension due to the adjacent edges. We introduce a passive tension $T_p$ on the new edge which balances this deficit

$$T_p = T_n - T_a = T_n - (m_{12} + m'_{12})$$

For example, if a perfectly symmetric quartet collapses when the vertex angle facing the collapsing edge is $90°$, $T_1 = T_2 = T'_1 = T'_2 = 1$ and $T_0 = T_n = \sqrt{2}$. Therefore, $T_{a,n} = 2 - \sqrt{2} \approx 0.6$ and $T_{p,n} = \sqrt{2} - (2 - \sqrt{2}) \approx 0.8$. Note that by the triangle inequality, for any convex quadrilateral with perimeter $P$ and diagonals $D_1, D_2$, one has $P/2 \leq D_1 + D_2 \leq P$. Applying this to the quadrilateral formed by the two tension triangles at the collapsing interface, we get $T_{a,n} \geq 0$ and $T_{p,n} \geq 0$: the handover formula always results in positive active and passive tensions. Furthermore, the 'handover' mechanism robustly generates irreversible T1s: if a junction were to collapse back after a T1, the newly formed junction would inherit high myosin levels and, therefore, be likely to collapse again.

The passive tension subsequently relaxes visco-elastically with rate $\tau_p^{-1}$. Combining this with the feedback equation yields *Equation 34*. The relaxation rate $\tau_p = \tau_T/6$ was fit to the measured tension decay post intercalation.

## Numerical implementation

We integrated *Equation 34* using a 4th order Runge-Kutta method as implemented in the SciPy software package (*Virtanen et al., 2020*), using as initial conditions the vertex angles in the experimental data at time $t = 5 > \min$ (the vertex angles from the data were temporally smoothed with a window of $2 > \min$ to reduce noise).

In our simulations, we triggered an intercalation event once the collapsing edge length, as determined by energy minimization, reached 0. We then initialized the passive tension as given by *Equation 37* and simulated the combined active/passive tension dynamics. The overall time and length scales were likewise fit to the data. The feedback exponent was $n = 4$. Numerically, energy minimization was carried out using the scipy.optimize package, specifically its Nelder-Mead optimizer.

## Passive intercalation

Within the same framework, we can also model passive intercalations, shown in *Figure 3F*. We can either specify cortical tension dynamics (in this case, tension homeostasis, $\dot{T} \sim -(T - 1)$) and then minimize an elastic energy that includes the influence of externally imposed shear. This will trivially reproduce the isogonal behavior shown in *Figure 2E*. Alternatively one can reason that on the dorsal side, there is very little active tension (cortical myosin), so cortical tensions are expected to be low. Then, the passive elastic energy, together with the imposed shear, should determine the also angles. Both models are consistent with the data.

In *Figure 3F*, we show simulations based on the latter option. We implement the externally imposed shear causing the passive intercalations as follows. We calculate the four cell centroids $\mathbf{c}_i$ of the cell quartets and demand that at simulation time $t$

$$\mathbf{c}_i(t) = S(t).\mathbf{c}_i(t = 0) \tag{38}$$

where $S(t)$ is a diagonal matrix representing (area-preserving) shear, increasing over time with constant shear rate $\gamma$.

We then minimize the elastic energy under the constraint (numerically, this is done by adding a penalty term to the elastic energy), but this time with respect to both the lengths $\ell_i$ and the angles $\phi_i$. As long as the shear modulus $\mu \ll \lambda$, this procedure results in approximately isogonal behavior. We chose $\mu = 0.01\lambda$. To obtain continuity across the intercalation, the rest shape $S_0$ needs to be adjusted. Otherwise, post intercalation, the cell quartet jumps back to a hexagonal lattice configuration. This can be done by introducing a viscous relaxation scale for the rest shape; for simplicity, we simply set the rest shape to the current shape tensor at the time of intercalation. The initial conditions are obtained by varying the angle between the shear axis and the collapsing interface orientation from $0° - 25°$.

## Appendix 6

### Role of the posterior midgut

To obtain additional evidence on the influence of posterior midgut (PMG) invagination on GBE, we carried out a quantitative re-analysis of data published in *Collinet et al., 2015*. In brief, in *Collinet et al., 2015* – among other experiments and analyses – the authors used an IR laser to fuse germ band tissue to the overlaying vitelline membrane by cauterization. This creates a 'cauterization fence' along the posterior end of the germ band which mechanically uncouples the germ band and the region of PMG invagination (*Collinet et al., 2015*). The authors then analyze tissue between the cauterization fence and the PMG. In wild-type embryos, they find that this tissue rips off from the cauterization fence, indicating pulling forces. This tissue ripping is also observed in mutants where force generation in the germ band is impaired (*eve/rnt Lefebvre et al., 2023* and *twi/snl Gustafson et al., 2022*; *Lefebvre et al., 2023*). In contrast, for *tor* mutants where the PMG does not invaginate, no tissue ripping is observed (*Collinet et al., 2015*). This suggests that the PMG exerts a pulling force on the germ band.

However, tissue ripping is only a qualitative measure for such pulling forces. To obtain a more quantitative estimate of the relative contributions of internally generated forces in the germ band and 'external' pulling forces from the PMG we re-analyzed the movies from *Collinet et al., 2015* using Particle Image Velocimetry (PIVlab *Thielicke and Sonntag, 2021*) to measure the average tissue flow in the region posterior to the cauterization fence. The results are shown in *Appendix 6—figure 1*. In *tor* embryos, where no PMG invagination occurs, there is essentially no tissue flow, in line with the qualitative assessment from tissue ripping. Since the PMG is attached to the vitelline membrane in the scab domain (*Münster et al., 2019*), it acts as a barrier in *tor* embryos, preventing the germ band from elongating. Notably, in the *eve/rnt* and the *twi/snl* embryo, tissue flow velocity posterior of the cauterized region is reduced by about two-fold compared to wild-type. This shows that impairing active force generation in the germ band has a strong influence on the tissue flow anterior of the PMG. This flow originates from the dorso-ventral contraction of the germ band that pulls on the dorsal amnioserosa and drives the hyperbolic flow there (*Streichan et al., 2018*). These quantitative observations based on the data from *Collinet et al., 2015* are in line with the much-reduced tissue flow in *eve* and *twist* embryos imaged *in toto* using light-sheet microscopy (*Lefebvre et al., 2023*; *Gustafson et al., 2022*).

The above tissue flow quantification is complemented by the following 'anatomical' considerations. As shown in *Bailles et al., 2019*, the tissue that will invaginate to form the posterior midgut consists of two distinct regions, the 'initiation' and the 'propagation' region, both have an AP-length of approximately 7–8 cell diameters. On the other hand, the germ band has a length of approximately 40 cell diameters (*Figure 1A*). Invagination of the initiation domain can hence create at most a strain of ca. $8/40 = 0.2$ (since the initiation region is anchored on its anterior side). Moreover, the PMG forms a deep fold separating its anterior front, where the force driving invagination originates (*Bailles et al., 2019*), from the germ band. It is unclear how a pulling force could be transmitted across this fold. Instead, the pulling force originating from the hyperbolic fixed point on the dorsal pole bypasses the PMG laterally where it manifests as isogonal stretching along the AP axis (see *Figure 6B'*).

Going forward, it would be interesting to image *tor* and *eve* embryos with a membrane-marker *in toto* and analyze them with the methods presented here. *In toto* imaging is crucial to resolve the contributions of different tissue regions as it allows to precisely time-align the recordings (*Mitchell et al., 2022*).

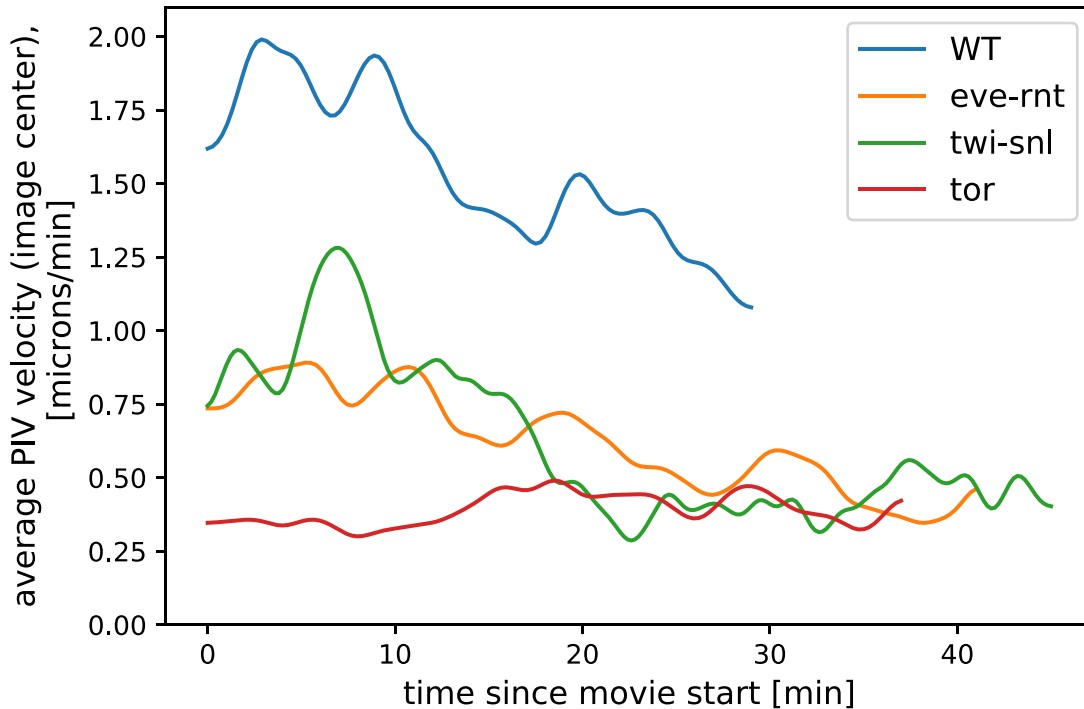

**Appendix 6—figure 1.** Particle Image Velocimetry (PIV)-analysis of tissue flow ahead of a DV cauterization fence in different genetic backgrounds. In the *eve/rnt* and the *twi/snl* embryo, where active force generation in the germ band is impaired, tissue flow is strongly reduced compared to wild-type (WT). For analysis, we excluded the cauterization fence itself. Data from *Collinet et al., 2015* ($N = 1$ movie for each genotype).

