## [Editor Report · eLife assessment]

This **important** study analyzes in an original way how tension pattern dynamics can reveal the contribution of active versus passive intercalation during tissue elongation. The authors develop a **compelling**, elegant analytical framework (isogonal tension decomposition) to disentangle the passive (adjacent tissues pulling) and active (local tension anisotropy) contributions to intercalation events. This allows the generation of global maps of tissue mechanics that will be extremely helpful in the field of biomechanics.

---

## [Referee Report · Reviewer #2 (Public review)]

Main comment from 1st review:

Weaknesses:

The modeling is interesting, with the integration of tension through tension triangulation around vertices and thus integrating force inference directly in the vertex model. However, the authors are not using it to test their hypothesis and support their analysis at the tissue level. Thus, although interesting, the analysis at the tissue level stays mainly descriptive.

Comments on the revised version:

My main concern was that the author did not use the analysis of mutant contexts such as Snail and Twist to confirm their predictions. They made a series of modifications, clarifying their conclusions. In particular, they now included an analysis of Snail mutant and show that isogonal deformations in the ventro-lateral regions are absent when the external pulling force of the VF is abolished, supporting the idea that isogonal strain could be used as an indicator of external forces (Fig7 and S6).

They further discuss their results in the context of what was published regarding the mutant backgrounds (fog, torso-like, scab, corkscrew, ksr) where midgut invagination is disrupted, and where germ band buckles, and propose that this supports the importance of internal versus external forces driving GBE.

Overall, these modifications, in addition to clarifications in the text, clearly strengthen the manuscript.

---

## [Referee Report · Reviewer #3 (Public review)]

In their article "The Geometric Basis of Epithelial Convergent

Extension", Brauns and colleagues present a physical analysis of *Drosophila* axis extension that couples in toto imaging of cell contours (previously published dataset), force inference, and theory. They seek to disentangle the respective contributions of active vs passive T1 transitions in the convergent extension of the lateral ectoderm (or germband) of the fly embryo.

The revision made by the authors has greatly improved their work, which was already very interesting, in particular the use of force inference throughout intercalation events to identify geometric signatures of active vs passive T1s, and the tension/isogonal decomposition. The new analysis of the Snail mutant adds a lot to the paper and makes their findings on the criteria for T1s very convincing.

About the tissue scale issues raised during the first round of review. Although I do not find the new arguments fully convincing (see below), the authors did put a lot of effort to discuss the role of the adjacent posterior midgut (PMG) on extension, which is already great. That will certainly provide the interested readers with enough material and references to dive into that question.

I still have some issues with the authors' interpretation on the role of the PMG, and on what actually drives the extension. Although it is clear that T1 events in the germ band are driven by active local tension anisotropy (which the authors show but was already well-established), it does not show that the tissue extension itself is powered by these active T1s. Their analysis of "fence" movies from Collinet et al 2015 (Tor mutants and Eve RNAi) is not fully convincing. Indeed, as the authors point out themselves, there is no flow in Tor mutant embryos, even though tension anisotropy is preserved. They argue that in Tor embryos the absence of PMG movement leaves no room for the germband to extend properly, thus impeding the flow. That suggests that the PMG acts as a barrier in Tor mutants - What is it attached to, then? The authors also argue that the posterior flow is reduced in "fenced" Eve RNAi embryos (which have less/no tension anisotropy), to justify their claim that it is the anisotropy that drives extension. However, previous data, including some of the authors' (Irvine and Wieschaus, 1994 - Fig 8), show that the first, rapid phase of germband extension is left completely unaffected in Eve mutants (that lack active tension anisotropy). Although intercalation in Eve mutants is not quantified in that reference, this was later done by others, showing that it is strongly reduced. Similarly, the Cyto-D phenotype from Clement et al 2017, in which intercalation is also strongly reduced, also displays normal extension.

---

## [Author Response]

The following is the authors’ response to the current reviews.

We thank the reviewers and editor for their positive assessment of our work. For the Version of Record, we have made small revisions addressing the remaining concerns of reviewer #3. We have also reformatted the supplementary material to conform to eLife’s style.

While the manuscript was under review, we discussed our work with Bill Bialek, who suggested clarifying the effect of cell rearrangements on genetic patterns. Using the tracked cell trajectories we found that the highly coordinated intercalations in the germ band preserve the relative AP positions of cells. We have added an Appendix subsection (Appendix 1.5) explaining this finding and highlighting its relevance in a short paragraph added to the discussion.

**Reviewer #2**
Main comment from 1st review:Weaknesses:The modeling is interesting, with the integration of tension through tension triangulation around vertices and thus integrating force inference directly in the vertex model. However, the authors are not using it to test their hypothesis and support their analysis at the tissue level. Thus, although interesting, the analysis at the tissue level stays mainly descriptive.Comments on the revised version:My main concern was that the author did not use the analysis of mutant contexts such as Snail and Twist to confirm their predictions. They made a series of modifications, clarifying their conclusions. In particular, they now included an analysis of Snail mutant and show that isogonal deformations in the ventro-lateral regions are absent when the external pulling force of the VF is abolished, supporting the idea that isogonal strain could be used as an indicator of external forces (Fig7 and S6).They further discuss their results in the context of what was published regarding the mutant backgrounds (fog, torso-like, scab, corkscrew, ksr) where midgut invagination is disrupted, and where germ band buckles, and propose that this supports the importance of internal versus external forces driving GBE.Overall, these modifications, in addition to clarifications in the text, clearly strengthen the manuscript.

We thank the reviewer for assessing our manuscript again and are happy to hear that they find the added data on the snail mutant convincing and that our revised manuscript is stronger.

**Reviewer #3**
In their article "The Geometric Basis of Epithelial Convergent Extension", Brauns and colleagues present a physical analysis of *Drosophila* axis extension that couples in toto imaging of cell contours (previously published dataset), force inference, and theory. They seek to disentangle the respective contributions of active vs passive T1 transitions in the convergent extension of the lateral ectoderm (or germband) of the fly embryo.The revision made by the authors has greatly improved their work, which was already very interesting, in particular the use of force inference throughout intercalation events to identify geometric signatures of active vs passive T1s, and the tension/isogonal decomposition. The new analysis of the Snail mutant adds a lot to the paper and makes their findings on the criteria for T1s very convincing.About the tissue scale issues raised during the first round of review. Although I do not find the new arguments fully convincing (see below), the authors did put a lot of effort to discuss the role of the adjacent posterior midgut (PMG) on extension, which is already great. That will certainly provide the interested readers with enough material and references to dive into that question.

We appreciate the referee’s positive assessment of our manuscript and their careful reading and constructive feedback. In particular, we are happy to hear that the referee finds our added data on the snail mutant very convincing and finds that the extended discussion on the role of the PMG is helpful. We address the remaining concerns in our detailed response below.

I still have some issues with the authors' interpretation on the role of the PMG, and on what actually drives the extension. Although it is clear that T1 events in the germ band are driven by active local tension anisotropy (which the authors show but was already well-established), it does not show that the tissue extension itself is powered by these active T1s. Their analysis of "fence" movies from Collinet et al 2015 (Tor mutants and Eve RNAi) is not fully convincing. Indeed, as the authors point out themselves, there is no flow in Tor mutant embryos, even though tension anisotropy is preserved. They argue that in Tor embryos the absence of PMG movement leaves no room for the germband to extend properly, thus impeding the flow. That suggests that the PMG acts as a barrier in Tor mutants - What is it attached to, then?

We thank the referee for pointing out this omission: The PMG is attached to the vitelline membrane in the scab domain (Munster et al. Nature 2019) and is also obstructed from moving by more anterior laying tissue (amnioserosa). It therefore acts as an obstacle for GBE extension if it fails to invaginate (e.g. in a Tor embryo). We have clarified this in the discussion of the Tor mutants.

The authors also argue that the posterior flow is reduced in "fenced" Eve RNAi embryos (which have less/no tension anisotropy), to justify their claim that it is the anisotropy that drives extension. However, previous data, including some of the authors' (Irvine and Wieschaus, 1994 - Fig 8), show that the first, rapid phase of germband extension is left completely unaffected in Eve mutants (that lack active tension anisotropy). Although intercalation in Eve mutants is not quantified in that reference, this was later done by others, showing that it is strongly reduced.

The quantification of GBE in Irvine and Wieschaus 1994 was based on the position of the PMG from bright field imaging, making it hard to distinguish the contributions of ventral furrow, PMG, and germ band, particularly during the early phase of GBE where all these processes happen simultaneously. More detailed quantifications based on PIV analysis of in toto light-sheet imaging show significantly reduced tissue flow in eve mutants after the completion of ventral furrow invagination (Lefebvre et al., eLife 2023). That the initial fast flow is driven by ventral furrow invagination, not by the PMG is apparent from twist/snail embryos where the initial phase is significantly slower (Lefebvre et al., eLife 2023, Gustafson et al., Nat Comms 2022). We have added these references to the re-analysis and discussion of the Collinet et al 2015 experiments.

Similarly, the Cyto-D phenotype from Clement et al 2017, in which intercalation is also strongly reduced, also displays normal extension.

We agree that a careful quantification of tissue flow in Cyto-D-treated embryos would be interesting. Whether they show normal extension is not clear from the Clement et al. 2017 paper, as no quantification of total tissue flow is performed and no statements regarding extension are made there.

**Reviewer #3 (Recommendations For The Authors):**
A lot of typos / grammar mistakes / repetitions are still found here and there in the paper. Authors should plan a careful re-reading prior to final publication.

We have carefully checked the manuscript and fixed the typos and grammar mistakes.

I failed to point to a very relevant reference in the previous round of review, which I think the authors should cite and comment: A review by Guirao & Bellaiche on the mechanics of intercalation in the fly germband, which notably discusses the passive/active andstress-relaxing/stress-generating nature of T1s. (Guirao and Bellaiche, Current opinions in cell biology 2017), in particular figures 1 and 2.

We thank the referee for pointing us to this relevant reference which we now cite in the introduction.

Any new arguments/discussion the authors see fit to include in the paper to comment on the Eve/Tor phenotypes. As far as I am concerned, I am not fully convinced at the moment (see review), but I think the paper has other great qualities and findings, and now (since the first round of review) sufficiently discusses that particular matter. I leave it up to the authors how much (more) they want to delve into this in their final version!

We have added clarifications and references to the discussion of the Eve/Tor phenotypes.

The following is the authors’ response to the original reviews.

**Public Review:**

**Joint Public Review:**
Summary:Brauns et al. work to decipher the respective contribution of active versus passive contributions to cell shape changes during germ band elongation. Using a novel quantification tool of local tension, their results suggest that epithelial convergent extension results from internal forces.

Reading this summary, and the eLife assessment, we realized that we failed to clearly communicate important aspects of our findings in the first version of our manuscript. We therefore decided to largely restructure and rewrite the abstract and introduction to emphasize that:

Our analysis method identifies active vs passive contributions to cell **and tissue** shape changes during epithelial convergent extension

In the context of *Drosophila* germ band extension, this analysis provides evidence for a major role for internal driving forces rather than external pulling force from neighboring tissue regions (posterior midgut), thus settling a question that has been debated due to apparently conflicting evidence from different experiments.

Our findings have important implications for local, bottom-up self-organization vs top-down genetic control of tissue behaviors during morphogenesis.

Strengths:The approach developed here, tension isogonal decomposition, is original and the authors made the demonstration that we can extract comprehensive data on tissue mechanics from this type of analysis.They present an elegant diagram that quantifies how active and passive forces interact to drive cell intercalations.The model qualitatively recapitulates the features of passive and active intercalation for a T1 event.Regions of high isogonal strains are consistent with the proximity of known active regions.

We think this statement is somewhat ambiguous and does not summarize our findings precisely. A more precise statement would be that high isogonal strain identifies regions of passive deformation, which is caused by adjacent active regions.

They define a parameter (the LTC parameter) which encompasses the geometry of the tension triangles and allows the authors to define a criterium for T1s to occur.The data are clearly presented, going from cellular scale to tissue scale, and integrating modeling approaches to complement the thoughtful description of tension patterns.Weaknesses:The modeling is interesting, with the integration of tension through tension triangulation around vertices and thus integrating force inference directly in the vertex model. However, the authors are not using it to test their hypothesis and support their analysis at the tissue level. Thus, although interesting, the analysis at the tissue level stays mainly descriptive.

We fully agree that a full tissue scale model is crucial to support the claims about tissue scale self-organization we make in the discussion. However, the full analysis of such a model is beyond the scope of the present manuscript. We have therefore split off that analysis into a companion manuscript (Claussen et al. 2023). In this paper, we show that the key results of the tissue-scale analysis of the *Drosophila* embryo, in particular the order-to-disorder transition associated with slowdown of tissue flow, are reproduced and rationalized by our model.

We now refer more closely to this companion paper to point the reader to the results presented there.

Major points:(1) The authors mention that from their analysis, they can predict what is the tension threshold required for intercalations in different conditions and predict that in Snail and Twist mutants the T1 tension threshold would be around √2. Since movies of these mutants are most probably available, it would be nice to confirm these predictions.

This is an excellent suggestion. We have included an analysis of a recording of a Snail mutant, which is presented in the new Figures 4 and S6. As predicted, we find that isogonal deformations in the ventro-lateral regions are absent when the external pulling force of the VF is abolished. Further, in the absence of isogonal deformation, T1 transitions indeed occur at a critical tension of approx. √2, as predicted by our model. Both of these results provide important experimental evidence for our model and for isogonal strain as a reliable indicator of external forces.

(2) While the formalism is very elegant and convincing, and also convincingly allows making sense of the data presented in the paper, it is not all that clear whether the claims are compatible with previous experimental observations. In particular, it has been reported in different papers (including Collinet et al NCB 2015, Clement et al Curr Biol 2017) that affecting the initial Myosin polarity or the rate of T1s does not affect tissue-scale convergent extension. Analysis/discussion of the Tor phenotype (no extension with myosin anisotropy) and the Eve/Runt phenotype (extension without Myosin anisotropy), which seem in contradiction with an extension mostly driven by myosin anisotropy.

We are happy to read that the referees find our approach elegant and convincing. The referees correctly point out that we have failed to clearly communicate how our findings connect to the existing literature on *Drosophila* GBE. Indeed, the conflicting results reported in the literature on what drives GBE – internal forces (myosin anisotropy) or external forces (pulling by the posterior midgut) – were a motivation for our study. We have extensively rewritten the introduction, results section (“Isogonal strain identifies regions of passive tissue deformation”), and discussion (“Internal and external contributions to germ band extension”) in response to the referee’s request.

In brief, distinguishing active internal vs passive external driving of tissue flow has been a fundamental open question in the literature on morphogenesis. Our tension-isogonal decomposition now provides a way to answer this question on the cell scale, by identifying regions of passive deformation due to external forces. As we now explain more clearly, our analysis shows that germ band extension is predominantly driven by internal tension dynamics, and not pulling forces from the posterior midgut.

We put this cell-scale evidence into the context of previous experimental observations on the tissue scale: Genetic mutants (fog, torso-like, scab, corkscrew, ksr), where posterior midgut invagination is disrupted (Muenster et al. 2019, Smits et al. 2023). In these mutants, the germ band buckles forming ectopic folds or twists into a corkscrew shape as it extends, pointing towards a buckling instability characteristic of internally driven extensile flows.

To address the apparently conflicting evidence from Collinet et al. 2015, we carried out a

quantitative re-analysis of the data presented in that reference (see new SI section 3 and Fig.S11). The results support the conclusion that the majority of GBE flow is driven internally, thus resolving the apparent conflict.

Lastly, as far as we understand, Clement et al. 2017 appears to be compatible with our picture of active T1 transitions. Clement et al. report that the actin cortex, when loaded by external forces, behaves visco-elastically with a relaxation time of the order of minutes, in line with our model for emerging interfaces post T1.

We again thank the referees for prompting us to address these important issues and believe that including their discussion has significantly strengthened our manuscript.

**Recommendations for the authors:**
Minor points:- Fig 2 : authors should state in the main text at which scale the inverse problem is solved. (Intercalating quartet, if I understood correctly from the methods) ? and they should explain and justify their choice (why not computing the inverse at a larger scale).

We have rephrased the first sentence of the section “Cell scale analysis” to clarify that we use local tension inference. This local inference is informative about the relative tension of one interface to its four neighbors. The focus on this local level is justified because we are interested in local cell behaviors, namely rearrangements. Tension inference is also most robust on the local level, since this is where force balance, the underlying physical determinant of the link between mechanics and geometry, resides. In global tension inference, spurious large scale gradients can appear when small deviations from local force balance accumulate over large distances. We have added a paragraph in SI Sec. 1.4 to explain these points.

-Fig 2 : how should one interpret that tension after passive intercalation (amnioserosa) is higher than before. On fig 2E, tension has not converged yet on the plot, what happens after 20 minutes ?

Recall that the inferred tension is the total tension on an interface. While on contracting interfaces, the majority of this tension will be actively generated by myosin motors, on extending interfaces there is also a contribution carried by passive crosslinkers. The passive tension can be effectively viewed as viscous dissipation on the elongating interface as crosslinkers turn over (Clement et al. 2017). Note that this passive tension is explicitly accounted for in the model presented in Fig. 5. Notably, it is crucial for the T1 process to resolve in a new extending junction. In the amnioserosa, the tension post T1 remains elevated because the amnioserosa is continually stretched by the convergence of the germ band. The tension hence does not necessarily converge back to 1. However, our estimates for the tension after 20 mins post T1 are very noisy because most of the T1s happen relatively late in the movie (past the 25 min mark) and therefore there are only a few T1s where we can track the post-T1 dynamics for more than 20 mins.

We have added a brief explanation of the high post-T1 tension at the end of the section entitled “Relative tension dynamics distinguishes active and passive intercalations”. Further, we have moved up the section describing the minimal model right after the analysis of the relative tension during intercalations. We believe that this helps the reader better understand these findings before moving on to the tension-isogonal decomposition which generalizes them to the tissue scale.

Page 7-8 / Figure 3: It is unclear how the decomposition into (1) physical shape (2) tension shape (2) isogonal shape works exactly. A more detailed explanation and more clear illustration of what a quartet is and its labels could help.

We have added a more detailed explanation in the main text. See our response to the longer question regarding this point below.

-What exactly defines the boundary curve in figure 3E? How is it computed?

We have added a sentence in the caption for Fig. 3E explaining that the boundary curve is found by solving Eq. (1) with l set to zero for the case of a symmetric quartet. We have also added a brief explanation immediately below Eq. (1) pointing out that this equation defines the T1 threshold in the space of local tensions T_i in terms of the isogonal length l_iso.

-The authors should consider incorporating some details described in the SI file to the main text to clarify some points, as long as the accessible style of the manuscript can be kept. The points mentioned below may also be clarified in the SI doc. The specific points that could be elaborated are: Page 7-8 / Figure 3: It is unclear how the decomposition into (1) physical shape (2) tension shape (2) isogonal shape works exactly. A more detailed explanation and more clear illustration of what a quartet is and its labels could help. The mapping to Maxwell-Cremona space is fine, but which subset is the quartet? For a set of 4 cells with two shared vertices and a junction, aren't there 5 different tension vectors? Are we talking two closed force triangles? Separately, how do you exactly decompose the deformation (of 4 full cell shapes or a subset?) into isogonal and non-isogonal parts? What is the least squares fit done over - is this system underdetermined? Is this statistically averaged or computed per quartet and then averaged?

We thank the referees for pointing us to unclear passages in our presentation. We hope that our revisions have resolved the referee’s questions. As described above, we have clarified the tension-isogonal decomposition in the main text. We have also revised the corresponding SI section (1.5) to address the above questions. A sketch of the quartet with labels is found in SI Fig. S7A which we now refer to explicitly in the main text.

We always consider force-balance configurations, i.e. closed force triangles. Therefore in the “kite” formed by two adjacent tension triangles, only three tension vectors are independent.

The decomposition of deformation is performed as follows: For each of the four cells, the center of mass c_i is calculated. Next, tension inference is performed to find the two tension triangles with tension vectors T_ij. Now there are three independent centroidal vectors c_j - c_i and three corresponding independent tension vectors T_ij. We define the isogonal deformation tensor I_quratet as the tensor that maps the centroidal vectors to the tension vectors. In general this is not possible exactly, because I_quartet has only three independent components, but there are six equations.

The plots in Fig. 3C, C’ are obtained by performing this decomposition for each intercalating quartet individually. The data is then aligned in time and ensemble averages are calculated for each timepoint.

For tissue-scale analysis in Fig. 6, the decomposition is performed for individual vertices (i.e. the corresponding centroidal and tension triangles) and then averaged locally to find the isogonal strain fields shown in Fig. 6B, B’.

- Line 468: "Therefore, tissue-scale anisotropy of active tension is central to drive and orient convergent-extension flow [10, 57, 59, 60]." Authors almost never mention the contribution of the PMG to tissue extension. Yet it is known to be crucial (convergent extension in Tor mutants is very much affected). Please discuss this point further.

The referees raise an important point: as discussed in our response to major point (2), we now explicitly discuss the role of internal (active tension) and external (PMG pulling) forces during germ band extension. Please see our response to major point (2) for the changes we made to the manuscript to address this.

In particular, we now explain that in mutants where PMG invagination is impaired (fog, torso-like, torso, scab, corkscrew), the germ band buckles out of plane or extends in a twisted, corkscrew fashion (Smits et al. 2023). This shows that the germ band generates extensile forces largely internally. In torso mutants, the now stationary PMG acts as a barrier which blocks GBE extension; the germ band buckles as a response.

The role of PMG invagination hence lies not in creating pulling forces to extend the germ band, but rather in “making room” to allow for its orderly extension. As shown by the genetics mutants just discussed, the synchronization of PMG invagination and GBE is crucial for successful gastrulation.

-Typos:Line 74: how are intercalations areLine 84: vertices verticesLine 233: very differentlyLine 236: are canLine 390: energy which is the isogonal mode mustLine 1585: reveals showLine 603: area Line 618: in terms of on the

We have fixed these typos.